# An Optimistic Algorithm for online CMDPS with Anytime Adversarial Constraints

**Jiahui Zhu** [1]  **Kihyun Yu** [2]  **Dabeen Lee** [2]  **Xin Liu** [3]  **Honghao Wei** [1]

## Abstract

Online safe reinforcement learning (RL) plays a key role in dynamic environments, with applications in autonomous driving, robotics, and cybersecurity. The objective is to learn optimal policies that maximize rewards while satisfying safety constraints modeled by constrained Markov decision processes (CMDPs). Existing methods achieve sublinear regret under stochastic constraints but often fail in adversarial settings, where constraints are unknown, time-varying, and potentially adversarially designed. In this paper, we propose the **O**ptimistic **M**irror **D**escent Primal-**D**ual (OMDPD) algorithm, the first to address online CMDPs with anytime adversarial constraints. OMDPD achieves optimal regret $\tilde{\mathcal{O}}(\sqrt{K})$ and strong constraint violation $\tilde{\mathcal{O}}(\sqrt{K})$ without relying on Slater's condition or the existence of a strictly known safe policy. We further show that access to accurate estimates of rewards and transitions can further improve these bounds. Our results offer practical guarantees for safe decision-making in adversarial environments.

## 1. Introduction

Online safe reinforcement learning (RL) has been applied successfully across various domains, including autonomous driving (Isele et al., 2018), recommender systems (Chow et al., 2017), and robotics (Achiam et al., 2017). It enables the efficient development of policies that adhere to essential safety requirements, such as collision avoidance, budget compliance, and reliability. In safe RL, the objective is to learn an optimal policy that maximizes cumulative rewards while satisfying *expected cumulative safety constraints* when interacting with an unknown environment. These safety-critical sequential decision-making problems are formally modeled as constrained Markov decision processes (CMDPs) (Altman, 1999). We study the problem of learning such policies in episodic CMDPs where the agent must balance exploration and exploitation to minimize **strong regret**(regret and hard constraint violation), a metric quantifying the cumulative performance loss relative to the optimal safe policy—while avoiding excessive constraint violations.

In many practical scenarios, especially in safety-critical applications, the environment is often dynamic, with conditions that can change unpredictably or even adversarially. Relying solely on stationary models can lead to suboptimal or unsafe outcomes, as these models fail to capture the time-varying nature of constraints and system dynamics. For instance, in autonomous driving (Kirschner et al., 2021), it is crucial to avoid collisions in variable environments influenced by changing traffic flows and weather conditions. Similarly, in cybersecurity (Yinka-Banjo & Ugot, 2020), adversarial CMDPs can model interactions between system defenders and attackers, requiring defenders to make decisions under uncertainty and strict system constraints while accounting for potential adversarial actions. This necessitates the consideration of adversarial settings, where constraints may be unknown, time-dependent, and potentially adversarially designed to challenge the learning process. Existing methods (Efroni et al., 2020; Müller et al., 2023; Germano et al., 2023; Müller et al., 2024; Kitamura et al., 2024) provide sublinear regret guarantees for stochastic constraints but struggle to generalize to such adversarial cases. The adversarial setting is inherently more challenging due to the dynamic and unpredictable nature of constraints, compounded by the assumption that error cancellation in constraint violations is not allowed. Adversarial CMDPs are thus crucial for handling dynamic environments, ensuring robust and safe decision-making in situations where conventional stochastic models fall short.

Constraint violation is usually used to theoretically evaluate the performance of the safety of a safe RL algorithm. One commonly used constraint violation evaluates the policies in the beverage sense such that it allows error cancellation, as defined by (Efroni et al., 2020), involves summing positive

[1]School of Electrical Engineering & Computer Science, Pullman, USA [2]Department of Industrial & Systems Engineering, KAIST, Daejeon, South Korea [3]School of Information Science & Technology, ShanghaiTech University, Shanghai, China. Correspondence to: Jiahui Zhu <jiahui.zhu@wsu.edu>.

*Proceedings of the $42^{st}$ International Conference on Machine Learning*, Vancouver, Canada. PMLR 267, 2025. Copyright 2025 by the author(s).

(unsafe) and negative (safe) constraint violations, ensuring a sublinear total constraint violation during learning. In this paper we consider a stronger notion of constraint violation, focusing exclusively on the sum of positive errors. To illustrate, consider a cost function $d(\pi_k)$, which equals $-1$ when the policy $\pi_k$ used in episode $k$ is safe, and $1$ when it is unsafe. If half of the policies over $K$ episodes are safe and the other half are unsafe, the weak constraint violation—permitting cancellation—results in $[\sum_{k=1}^{K} d(\pi_k)]^+ = 0$, where $[\cdot]^+ = \max\{\cdot, 0\}$. However, under strong constraint violation, which disallows cancellation, the total violation becomes $\sum_{k=1}^{K} [d(\pi_k)]^+ = K/2$. Clearly, weaker sublinear constraint violations do not ensure relatively safe policies during learning.

In this work, we aim to address two fundamental research questions: **RQ1:** Can we design a unified algorithm that achieves the optimal order of regret and hard constraint violation in unknown CMDPs with both stochastic and adversarial costs under minimum assumption? **RQ2:** What are the bottlenecks for further improving the bound?

CMDPs with cumulative constraints that allow cancellation have been extensively studied under both model-free (Wei et al., 2022a;b; 2023; Ghosh et al., 2022; Bai et al., 2022) and model-based approaches (Ding et al., 2021; Liu et al., 2021a; Bura et al., 2021; Singh et al., 2020; Ding et al., 2021; Chen et al., 2022; Efroni et al., 2020). The study by (Qiu et al., 2020; Stradi et al., 2024a) focuses on CMDPs with only an adversarial reward function. Recent work (Germano et al., 2023; Stradi et al., 2024c) considers online learning CMDPs under strong constraint violation for the long-term average cost and sublinear regret and violation results are well established. However, we note that in our scenario, the constraints are sufficiently strict—particularly in adversarial settings—that *an average safe policy fails to guarantee safety for each individual episode*. Consequently, focusing on the long-term average constraint alone is less meaningful under these conditions. Other works such as (Ding & Lavaei, 2022; Wei et al., 2023) consider scenarios where rewards, costs, and transition kernels are non-stationary, assuming bounded total variation. However, all of the above works we mentioned are not applicable to settings with adversarial costs, and they only address weak constraint violations.

To address the aforementioned challenges, we propose the **O**ptimistic **M**irror **D**escent **P**rimal-**D**ual (OMDPD) algorithm that ensures optimal regret and strong constraint violation bounds with respect to the number of episodes $K$, regardless of whether the reward and cost functions are generated stochastically or adversarially. Our contributions are summarized as follows:

- We present the **first** work addressing online CMDPs with anytime adversarial constraints. Our work advances the theoretical understanding of CMDPs under unknown ad-

versarial cost functions by proposing a novel unified algorithm, OMDPD, capable of handling both stochastic and adversarial rewards/costs without relying on Slater's condition. OMDPD achieves $\tilde{\mathcal{O}}(\sqrt{K})$ regret and $\tilde{\mathcal{O}}(\sqrt{K})$ strong constraint violation when rewards and costs are either stochastic or adversarial, both of which are optimal with respect to the total number of learning episodes $K$.

- It is well known that one of the bottlenecks of forbidding algorithms for online CMDP from achieving a higher bound is because of the estimation errors of reward/cost and transition kernels. We further show that if a perfect simulator (generative model) is given such that we can have an accurate estimate of the reward and transition kernels (cost function is also not known and can be adversarial), our regret bound can be further improved to $\mathcal{O}(1)$ when the reward function(also unknown) is fixed.

## 2. More Related Work

Müller et al. (2023) proposes an augmented Lagrangian method for addressing CMDPs with strong constraint violations under a requirement of a strictly known safe policy. Stradi et al. (2024c) propose a primal-dual algorithm (CPD-PO), building on the policy optimization framework of (Luo et al., 2021), which achieves $\tilde{\mathcal{O}}(\sqrt{K})$ regret. However, neither of these works addresses the adversarial cost setting. In addition, Stradi et al. (2024b) consider the adversarial reward setting but still assume stochastic constraints, requiring strong assumptions such as access to a strictly feasible policy and knowledge of its associated cost. Clearly, Stradi et al. (2024b) also cannot be applied to adversarial constraint scenarios. Additional studies by (Müller et al., 2024) and (Kitamura et al., 2024) focus on last-iterate convergence under stochastic constraints, achieving rates of $\tilde{\mathcal{O}}(K^{0.93})$ and $\tilde{\mathcal{O}}(K^{6/7})$, respectively. These results crucially rely on a stationary setting. A detailed comparison of the theoretical results between our algorithm and the most existing studies is summarized in Table 1.

## 3. Preliminaries

**Notation.** For any $n \in \mathbb{N}$, we use the short-hand notation $[n]$ to refer to the set of integers $\{1, \ldots, n\}$. For $x \in \mathbb{R}$, we define the operation $[x]^+ := \max\{0, x\}$ to be the positive truncation of $x$. Throughout the paper, we use $\|\cdot\|$ to denote the Euclidean norm. Additionally, for a given 1-strongly convex function $U$, we define the Bregman divergence between two points: $\mathcal{D}(a, b) = U(a) - U(b) - \langle \nabla U(a), a - b \rangle$.

We consider a finite-horizon episodic CMDP, which is defined as a tuple $\mathcal{M} = (\mu, \mathcal{S}, \mathcal{A}, H, \{\mathbb{P}_h\}_{h=1}^{H}, \{r_k\}_{k=1}^{K}, \{d_k\}_{k=1}^{K})$, where $\mu$ is the initial state distribution, $\mathcal{S}$ and $\mathcal{A}$ are the state and action spaces. We assume that both the state space and action space are finite and countable

| Algorithm | Regret | Adversarial Violation | Stochastic Violation | Slater's Condition | Known Safe Policy |
|---|---|---|---|---|---|
| (Efroni et al., 2020) | $\mathcal{O}(\sqrt{K})$ | N/A | $\mathcal{O}(\sqrt{K})$ | ✓ | No |
| (Müller et al., 2023)[†] | $\mathcal{O}(\sqrt{K})$ | N/A | $\mathcal{O}(\sqrt{K})$ | ✓ | Yes |
| (Stradi et al., 2024c)[†] | $\tilde{\mathcal{O}}(\sqrt{K})$ | N/A | $\tilde{\mathcal{O}}(\sqrt{K})$ | ✓ | No |
| (Müller et al., 2024)[†] | $\tilde{\mathcal{O}}(K^{0.93})$ | N/A | $\tilde{\mathcal{O}}(K^{0.93})$ | ✓ | No |
| (Kitamura et al., 2024)[†] | $\tilde{\mathcal{O}}(K^{\frac{6}{7}})$ | N/A | $\tilde{\mathcal{O}}(K^{\frac{6}{7}})$ | ✓ | No |
| **OMDPD**[†] | $\tilde{\mathcal{O}}(\sqrt{K})$ | $\tilde{\mathcal{O}}(\sqrt{K})$ | $\tilde{\mathcal{O}}(\sqrt{K})$ | ✗ | No |

*Table 1.* Comparison between OMDPD and existing related work. We omit the dependence on the dimension of the action, state space, and the number of steps in CMDPs here. † : Consider the stronger notion of constraint violation, which disallows cancellation. (Efroni et al., 2020) only consider a weaker version. (Müller et al., 2024) need to access a strictly feasible policy. More discussions can be found in Section 2.

with cardinalities $|\mathcal{S}| = S, |\mathcal{A}| = A$. In the online learning under finite-horizon episodic CMDPs, each episode $k \in [K]$ has $H$ steps and at each $h \in [H]$, we use $\mathbb{P}_h(s'|s, a) : \mathcal{S} \times \mathcal{A} \times \mathcal{S} \to [0, 1]$ to denote the transition kernel from state action pair $(s, a)$ to a next state $s'$ at step $h$. Without loss of generality, we assume that the reward function $\{r_k\}_{k=1}^K$ is a sequence of vectors at each episode $k \in [K]$, in particular, $r_k = (r_{k,1}, \ldots, r_{k,H})$, where $r_{k,h} : \mathcal{S} \times \mathcal{A} \to [0, 1], \forall h \in [H], k \in [K]$. Similar, the cost function $d_{k,h}$ at step $h$ in episode $k$ is $d_{k,h} : \mathcal{S} \times A \to [-1, 1]$, both rewards and costs are bounded for any $h \in [H], k \in [K]$. The transition kernels, reward functions, and cost functions are unknown. In this paper, we consider the *stochastic reward* where $r_k$ is a random variable distributed according to a distribution $R$ for every $k \in K$, with two different types of cost functions: *stochastic constraint* and *adversarial constraint*:

- **Stochastic cost:** In stochastic cost setting, $d_k$ is a random variable distributed according to a fixed probability distribution $D$ for every $k \in [K]$.

- **Adversarial cost:** In adversarial cost setting, $d_k$ are adversarially-selected and unknown.

In online CMDPs, the agent interacts with the CMDP by executing a policy $\pi = \{\pi_1, \pi_2, \ldots, \pi_H\}$, where $\pi_h(\cdot|s) \in \Delta(\mathcal{A})$, and $\Delta(\cdot)$ is a probability simplex. We denote by $\pi(\cdot|s)$ the probability distribution for a state $s \in \mathcal{S}$, Whenever the agent takes an action $a$ in state $s$ at step $h$, in episode $k$, it observes reward $r_{k,h}(s, a)$ sampled from a fixed distribution, and cost , $d_{k,h}(s, a)$ sampled either from a fixed distribution for the stochastic setting or chosen by an adversary for the adversarial setting. Then the value function for the reward and cost under the policy $\pi$ and transition kernel $p$ are defined as:

$$V^\pi(r_k, p) := \mathbb{E}\left[\sum_{h=1}^H r_{k,h}(s_h, a_h)|s_1, \pi, p\right] \quad (1)$$

$$V^\pi(d_k, p) := \mathbb{E}\left[\sum_{h=1}^H d_{k,h}(s_h, a_h)|s_1, \pi, p\right]. \quad (2)$$

In the following, we denote by $\Pi$ the set of all the possible policies the agent can choose from. we are interested in solving the following optimization problem:

$$\pi^* \in \arg\max_{\pi \in \Pi} V^\pi(\bar{r}, p)$$

$$s.t. \quad V^\pi(\bar{d}, p) \leqslant 0, \quad \textbf{(stochastic cost)}, \quad (3)$$

$$s.t. \quad V^\pi(d_k, p) \leqslant 0, \forall k \in [K] \quad \textbf{(adversarial cost)},$$

where $\bar{r} := \mathbb{E}_{r \sim R}[r], \bar{d} := \mathbb{E}_{d \sim D}[d]$. The solution of this offline optimization problem (3) is considered as the baseline algorithm which serves to evaluate the performances of online algorithms. The goal of the online CMDP problem is to learn an optimal policy to minimize cumulative regret and strong cumulative violation of constraints after $K$ episodes, which are defined below:

$$\text{Regret}(K) = \sum_{k=1}^K \left[V^{\pi^*}(\bar{r}, p) - V^{\pi_k}(\bar{r}, p)\right] \quad (4)$$

$$\text{Violation}(K) = \sum_{k=1}^K \left[V^{\pi_k}(\bar{d}, p)\right]^+, \textbf{(stochastic cost)} \quad (5)$$

$$\text{Violation}(K) = \sum_{k=1}^K \left[V^{\pi_k}(d_k, p)\right]^+, \textbf{(adversarial cost)} \quad (6)$$

Alternatively, the online optimization problem (3) can also be represented using the notion of occupancy measure (Altman, 1999) $\{q_h^\pi(s, a; p)\}_{h=1}^H$ under a policy $\pi$ and transition kernel $p$. For every $s \in \mathcal{S}, a \in \mathcal{A}$, we have the occupancy measure defined as:

$$q_h^\pi(s, a, s') = \Pr(s_{h+1} = s', s_h = s, a_h = a|p, \pi, s_1)$$

$$q_h^\pi(s, a) = \sum_{s' \in \mathcal{S}} q_h^\pi(s, a, s') \quad (7)$$

It is well known that the CMDP problem can be formulated as an LP problem (Altman, 1999), then the optimal occupancy measure can be obtained by solving the following optimization problem:

$$\max_{q \in \mathcal{Q}} \bar{r}^\top q \quad (8)$$

$$s.t. \quad \bar{d}^\top q \leqslant 0, \textbf{(stochastic cost)} \quad (9)$$

$$s.t. \ d_k^\top q \leqslant 0, \forall k \in [K], \textbf{(adversarial cost)}, \quad (10)$$

where $q \in [0,1]^{SAH}$ is the occupancy measure vector, with its values defined in Eq. (7), and $\mathcal{Q}$ represents the set of all valid occupancy measures. $\bar{r} \in [0,1]^{SAH}$ and $\bar{d}(d_k) \in [-1,1]^{SAH}$ denote the reward and cost vectors, respectively, with a slight abuse of notation. On either hand, for any $q$, the corresponding policy can be reconstructed as:

$$\pi_h^q(a|s) = \frac{q_h(s,a)}{\sum_{a'} q_h(s,a')}. \quad (11)$$

Given all the notations above, we denote the optimal solution to the optimization problem (3) as $q^*$, which also serves as the baseline. In this paper, following the standard assumptions, we consider the bandit feedback setting in which the learner observes only the rewards and costs for the chosen actions in the stochastic cost setting. However, in the adversarial cost setting, the full cost vector $d_k$ is revealed after episode $k$, while the reward remains as bandit feedback throughout.

# 4. Main Algorithm

In this section, we introduce our main algorithm and the designs behind that to ensure the optimal order on the regret and violation bounds.

## 4.1. Optimistic Estimates

To encourage the exploration of the unknown CMDP, we first need to use the principle of optimistic estimate (Auer et al., 2008). Let $n_h^{k-1}(s,a) = \sum_{k'=1}^{k-1} \mathbb{1}_{\{s_h^{k'}=s, a_h^{k'}=a\}}$ denote the number of times that the state-action pair $(s,a)$ is visited at step $h$ before episode $k$. Here, $(s_h^{k'}, a_h^{k'})$ denotes the state-action pair visited at step $h$ in episode $k'$, and $\mathbb{1}_{\{\cdot\}}$ is the indicator function. Then the empirical transition kernels, rewards and violations can be calculated as follows:

$$\hat{p}_h^{k-1}(s'|s,a) := \frac{\sum_{k'=1}^{k-1} \mathbb{1}_{\{s_h^{k'}=s, a_h^{k'}=a, s_{h+1}^{k'}=s'\}}}{n_h^{k-1}(s,a) \vee 1}, \quad (12)$$

$$\hat{r}_h^{k-1}(s'|s,a) := \frac{\sum_{k'=1}^{k-1} R_h^{k'}(s,a) \mathbb{1}_{\{s_h^{k'}=s, a_h^{k'}=a\}}}{n_h^{k-1}(s,a) \vee 1}, \quad (13)$$

$$\hat{d}_h^{k-1}(s'|s,a) := \frac{\sum_{k'=1}^{k-1} D_h^{k'}(s,a) \mathbb{1}_{\{s_h^{k'}=s, a_h^{k'}=a\}}}{n_h^{k-1}(s,a) \vee 1}, \quad (14)$$

where $a \vee b := \max\{a,b\}$. Remark that Eq. (14) is only used for the stochastic cost case. Then we can define the optimistic rewards, costs, and confidence set of transition kernels $B_{k,h}(s,a)$ as

$$\tilde{r}_{k,h}(s,a) := \hat{r}_h^{k-1}(s,a) + \beta_{k,h}^r(s,a) \quad (15)$$

$$\tilde{d}_{k,h}(s,a) := \hat{d}_h^{k-1}(s,a) - \beta_{k,h}^d(s,a) \quad (16)$$

$$B_{k,h}(s,a) := \{\tilde{p}_h(\cdot|s,a) \in \Delta(S) \mid \forall s' \in S:$$

$$|\tilde{p}_h(s'|s,a) - \hat{p}_h^{k-1}(s'|s,a)| \leqslant \beta_{k,h}^p(s,a,s')\}$$

$$B_k := \{\tilde{p} \mid \forall s,a,h : \tilde{p}_h(\cdot|s,a) \in B_{k,h}(s,a)\} \quad (17)$$

where $\beta_{k,h}^p(s,a,s') > 0$ is a UCB-type bonus, which denotes the confidence threshold for the transitions and is defined in Appendix B.1. Thus, we can construct a candidate set for selecting the policy at each episode $k$ as:

$$\mathcal{Q}_k := \{q^\pi(p) \in \mathbb{R}^{SAH} | \pi \in \Pi, p \in B_k\}, \quad (18)$$

where we denote $q^\pi(p) \in \mathbb{R}^{SAH}$ as the stacked occupancy measure vector under transition kernel $p$, a policy $\pi$, and $\Pi$ is the set of all the feasible policies.

## 4.2. Surrogate Objective Function

The objective of online CMDP learning is twofold: (1) to control the constraint violations over time, and (2) to maximize cumulative reward. Thus, after constructing the feasible candidate set for the policy $\mathcal{Q}_k$, our algorithm aims to solve the following optimization problem at each episode:

$$\max_{q \in \mathcal{Q}_k, d_k^\top q \leqslant 0} r_k^\top q. \quad (19)$$

Inspired by (Sinha & Vaze, 2024; Guo et al., 2022) we consider the following surrogate objective function with an exponential potential Lyapunov function: $\Phi(x) := \exp(\beta x) - 1$, for some constant $\beta > 0$:

$$f_k(q) = \alpha\left(-\tilde{r}_k^\top q + \Phi'(\lambda_k)[\tilde{d}_k^\top q]^+\right) - \frac{1}{2}\|q - q_k\|^2. \quad (20)$$

Using the prediction (estimation) of the reward and cost function together with the online mirror descent optimization methods we can achieve a tighter bound.

The dual variable $\lambda_k$ aims to track cumulative constraint violations during learning. Then through analysis of the drift term $\Phi(\lambda_k) - \Phi(\lambda_{k-1})$, the algorithm adaptively regulates long-term violation behavior: exponential growth in $\Phi(\lambda_k)$ dynamically amplifies constraint penalties during high-violation regimes, while bounded drift guarantees violations remain controllable. Together, these components enforce an safe exploration.

Specifically, we define the dual variable update as

$$\lambda_k = \lambda_{k-1} + \alpha[\tilde{d}_k^\top q_k]^+, \quad (21)$$

where $\tilde{d}_k$ is the estimated constraint vector (Eq. 16) for the stochastic cost and is replaced with $d_k$ when the cost is adversarial. The operator $[\cdot]^+$ considers only the positively violated cost to efficiently control the hard constraint. We then employ the Lyapunov function $\Phi(\lambda_k)$ to track the evolution of these violations. Then we will show later that the one-step Lyapunov drift can be bounded by:

$$\Phi(\lambda_k) - \Phi(\lambda_{k-1}) \leqslant \Phi'(\lambda_k) \cdot \alpha[\tilde{d}_k q_k]^+. \quad (22)$$

Then by using the drift-plus-penalty framework (Neely, 2010), we are able to minimize the surrogate cost functions $\{f_k(q)\}_{t=1}^T$ which is the combination of the drift upper bound (Eq. 22) and the cost function. More precisely, by selecting $q^*$ to minimize $f_k(q)$ within the feasible set $\mathcal{Q}_k$ and summing over $K$ episodes, we can obtain

$$\Phi(\lambda_K) + \alpha \sum_{k=1}^K \left(\tilde{r}_k^\top q^* - \tilde{r}_k^\top q_k\right) \leqslant \underbrace{\sum_{k=1}^K \left(f_k(q_k) - f_k(q^*)\right)}_{(I)} \quad (23)$$

We refer the term $(I)$ as $Regret_{alg}$. Hence, minimizing this algorithm's regret is crucial to bound the cumulative reward and constraint violation.

### 4.3. Optimistic OMD

To obtain a tight bound on term $(I)$, we adopt the Optimistic Online Mirror Descent (OMD) algorithm to dynamically control the constraints and adapt to evolving environments. The algorithm alternates between two phases in each iteration. In the *optimistic phase*, the algorithm constructs an anticipated occupancy measure, $\hat{q}_{k+1}$, by solving the following regularized optimization problem:

$$\hat{q}_{k+1} = \arg\min_{q \in \mathcal{Q}_k} \eta_k \langle q, \nabla f_k(q_k) \rangle + \mathcal{D}(q, \hat{q}_k),$$

where $\eta_k$ is the learning rate, and $\mathcal{D}(q, \hat{q}_k)$ represents the Bregman divergence, ensuring smooth updates. This step predicts the next occupancy measure by incorporating the gradient of the current potential function and regularization. In the *refinement phase*, the algorithm updates its policy by leveraging the predicted gradient $\nabla \hat{f}_{k+1}(\hat{q}_{k+1})$. Following the setup in (Rakhlin & Sridharan, 2013), we assume $\hat{f}_{k+1} = f_k$. The subsequent occupancy measure, $q_{k+1}$, is obtained by solving:

$$q_{k+1} = \arg\min_{q \in \mathcal{Q}_k} \eta_{k+1} \langle q, \nabla \hat{f}_{k+1}(\hat{q}_{k+1}) \rangle + \mathcal{D}(q, \hat{q}_{k+1}).$$

After we obtain the $q_{k+1}$, we can then construct $\pi_{k+1}$ using Eq. (11) and execute the policy and get estimations of the reward function, transition kernels, and cost functions if for the stochastic setting. The optimistic update is critical for enabling a tighter bound by incorporating historical gradients and occupancy measures. The full algorithm is presented in Algorithm 1.

## 5. Main Result

We first provide the main theoretical results of OMDPD.

**Theorem 5.1.** *Choose* $\alpha = \frac{1}{2(1+\sqrt{L_\delta})SAH}, \beta = \frac{SAH}{8\sqrt{\mathcal{C}}\sqrt{6SAHK}}$ *and denote* $\mathcal{C} = \sup_{q_1,q_2 \in Q} \mathcal{D}(q_1, q_2)$, *where* $L_\delta$ *is defined in Appendix B.1. Let* $\nabla_k = \nabla f_k(q_k), \nabla_{k-1} = \nabla f_{k-1}(q_{k-1})$ *and consider* $\eta_k =$

---

**Algorithm 1** OMDPD

**Input:** $q_1, \hat{q}_1 \in \mathcal{Q}_1, \tilde{r}_1 = \tilde{d}_1 = \lambda_1 = 0$, learning rate $\eta_k$.
**Parameters:** $\Phi(x) = \exp(\beta x) - 1, \alpha = \frac{1}{2(1+\sqrt{L_\delta})SAH}$,
$L_\delta$ is defined in Appendix B.1.
Define function $f_k$ by:

$$f_k(q) = \alpha(-\tilde{r}_k^\top q + \Phi'(\lambda_k)[\tilde{d}_k^\top q]^+) - \frac{1}{2}\|q - q_k\|^2$$

**for** $k = 1$ to $K$ **do**
 Construct the optimistic occupancy measure $\hat{q}_{k+1}$ by:

$$\hat{q}_{k+1} = \arg\min_{q \in \mathcal{Q}_k} \eta_k \langle q, \nabla f_k(q_k) \rangle + \mathcal{D}(q, \hat{q}_k)$$

 Assume $\hat{f}_{k+1} = f_k$; Compute $\eta_{k+1}$ and update $q_{k+1}$ by:

$$q_{k+1} = \arg\min_{q \in \mathcal{Q}_k} \eta_{k+1} \langle q, \nabla \hat{f}_{k+1}(\hat{q}_{k+1}) \rangle + \mathcal{D}(q, \hat{q}_{k+1})$$

 Construct $\pi_{k+1}$ by $q_{k+1}$ and execute policy, and get estimation $\tilde{r}_{k+1}, \tilde{d}_{k+1}$ by Eq.(15), (16).
 $d_{k+1}$ is revealed to the agent for the adversarial case.
 Update $\lambda_{k+1}$ as follows:

$$\lambda_{k+1} = \begin{cases} \lambda_k + \alpha[\tilde{d}_{k+1} q_{k+1}]^+ \text{(Stochastic Case)} \\ \lambda_k + \alpha[d_{k+1} q_{k+1}]^+ \text{(Adversarial Case)} \end{cases}$$

 Update set $\mathcal{Q}_{k+1}$ by Eq.(17).
**end for**
**Return:** $\pi_{K+1}$

---

$\sqrt{\mathcal{C}} \min\left\{ \frac{1}{\sqrt{\sum_{i=1}^{k-1}\|\nabla_i - \nabla_{i-1}\|_2^2} + \sqrt{\sum_{i=1}^{k-2}\|\nabla_i - \nabla_{i-1}\|_2^2}}, 1 \right\}$. *We have with probability at least* $1 - 2\delta$, *OMDPD achieves:*

$$Regret(K) \leqslant \tilde{\mathcal{O}}\left(\sqrt{\mathcal{N}SAH^3K} + S^2AH^3 + \sqrt{\mathcal{C}}\sqrt{SAHK} + SAH\right),$$

$$Violation(K) \leqslant \tilde{\mathcal{O}}\left(\sqrt{\mathcal{N}SAH^3K} + S^2AH^3 + \sqrt{\mathcal{C}}\sqrt{SAHK}\right)(Both\ settings)$$

Theorem 5.1 establishes the optimal $\tilde{\mathcal{O}}(\sqrt{K})$ regret and constraint violation bounds under minimum assumptions. This is the first result with optimal order in terms of the total episode $K$, for the online CMDPS with anytime adversarial constraints. Our results do not rely on the satisfaction of Slater's condition—a common assumption requiring the existence of a strictly feasible solution. Since in the adversarial setting, it is possible to make the slackness arbitrarily small by the adversary, thus the upper bound could be extremely large. The removal of this restrictive assumption represents a key theoretical contribution, as it aligns the algorithmic framework more closely with practical settings where Slater's condition cannot be ensured a priori.

*Remark* 5.2. Our approach depends on the cumulative vari-

ation of consecutive gradients, which is often very small when the reward is *fixed and known* or can be accurately estimated. Specifically, by choosing $\beta \leqslant \frac{2SAH}{3.5\sqrt{C}\sqrt{2SAHK}}$ and setting $\alpha, \eta_k$ as in Theorem 5.1, we can ensure that $\sum_{k=1}^{K}\left[\tilde{r}_k^\top q^* - \tilde{r}_k^\top q_k\right]$ is bounded by $\mathcal{O}(1)$. Consequently, in a CMDP with a generative model or a perfect simulator is given, such that the transitional kernels and reward functions can be accurately estimated(which eliminates the estimation step in Section 4.1, the error term $\tilde{\mathcal{O}}(\sqrt{\mathcal{N}SAH^3K})$) will be replaced by a constant that independent with episode $K$. Thus, the regret is bounded as $\mathcal{O}(1)$. The detailed proof is deferred in Appendix D.2.

*Remark* 5.3 (The Constant Bound $\mathcal{C}$). The constant $\mathcal{C}$ depends on the divergence measurement. If $\mathcal{D}$ is chosen as the KL divergence, it does not admit a uniform upper bound over the simplex, as $\mathrm{KL}(q\|q')$ may go to infinity. However, a smoothing track can be applied to keep all updated distributions bounded away from the boundary of the simplex. Such smoothing can ensure a boundness of $\mathcal{C}$ which is independent of the time horizon $K$. This technique is standard in online convex optimization under entropy regularization, more details can be found in (Wei et al., 2020).

### 5.1. Sketch of the Theoretical Analysis

In this section, we show the theoretical analysis of Algorithm 1. We first introduce the following facts for the CMDPs considered in this paper.

*Fact* 5.4. For any $q_1, q_2 \in \mathcal{Q}_k, \forall k \in [K]$, we have $\|q_1 - q_2\| \leqslant \sqrt{SAH}$.

*Fact* 5.5. For any $\tilde{r}_k, d_k$ or $\tilde{d}_k$, the reward/cost value function in terms of $q \in \mathcal{Q}_k$ is convex and Lipschitz continuous such that $|\tilde{r}_k^\top q_1 - \tilde{r}_k^\top q_2| \leqslant (1 + \sqrt{L_\delta})\sqrt{SAH}\|q_1 - q_2\|, |d_k^\top q_1 - d_k^\top q_2| \leqslant \sqrt{SAH}\|q_1 - q_2\|, |\tilde{d}_k^\top q_1 - \tilde{d}_k^\top q_2| \leqslant (1 + \sqrt{L_\delta})\sqrt{SAH}\|q_1 - q_2\|, \forall q_1, q_2 \in \mathcal{Q}_k, \forall k$, where $L_\delta$ is the logarithmic term defined in Appendix B.1.

Now, we introduce the **Good event** which captures the confidence of the current estimation and will be used to prove the policy used by OMDPD is comparable to the optimal solution. Our goal is to show that with a high probability, the true transition kernel lies in our confidence set such that the optimal solution is a feasible solution given the current estimation. We first show that **Good event** happens with a high probability. The detailed proof is deferred to Appendix B.1 due to page limit.

**Lemma 5.6.** *With probability at least $1 - \delta$, $\Pr[G] \geqslant 1 - \delta$, where $G$ is the good event and $\delta \in (0, 1)$.*

Basically, the good event shows that our estimation for the CMDP model is close to the true underlying model with high probability under the UCB-type exploration. Next, we will show that under the condition of **Good event**, the optimal solution of the CMDP problem Eq.(3) is a feasible

policy for the given $\mathcal{Q}_k$ in each episode $k \in [K]$, which make it possible to bound the regret and violation. Detailed proof can be found in Appendix B.1.

**Lemma 5.7.** *Conditioning on the **Good event** $G$, the optimal policy $\pi^*$ is a feasible solution policy for any episode $k \in [K]$ such that:*

$$\pi^* \in \left\{\pi : \tilde{d}_k^\top q^\pi(p') \leqslant 0, p' \in B_k\right\}$$

Therefore, $\pi^*$ is a **feasible solution** for any episode $k \in [K]$, where $q^{\pi^*}$ is the occupancy measure under the optimal policy $\pi^*$ and we denote $q^{\pi^*}$ by $q^*$ for simplicity. Lemma 5.7 ensures that the optimal solution $q^*$ is a feasible solution given the confidence set at any episode $k$, which makes it comparable to the policy used by OMDPD.

**Upper Bound of $Regret_{alg}$. (Eq. (23))** Based on the **Good event** $G$, we first present the upper bound of the $Regret_{alg}$. Using optimistic online mirror for selecting the policies in Algorithm 1, we have

**Lemma 5.8.** *Let $\mathcal{C} = \sup_{q_1,q_2 \in Q} \mathcal{D}_R(q_1, q_2)$, $\nabla_k = \nabla(f_k(q_k))$, $\nabla_{k-1} = \nabla(f_{k-1}(q_{k-1}))$, and define the learning rate as: $\eta_k = \sqrt{\mathcal{C}}\min\left\{\frac{1}{\sqrt{\sum_{i=1}^{k-1}\|\nabla_i - \nabla_{i-1}\|_2^2} + \sqrt{\sum_{i=1}^{k-2}\|\nabla_i - \nabla_{i-1}\|_2^2}}, 1\right\}$. Then, the regret is bounded as:*

$$Regret_{alg} \leqslant 3.5\sqrt{\mathcal{C}}\left(\sqrt{\sum_{k=1}^{K}\|\nabla_k - \nabla_{k-1}\|_2^2 + 1}\right)$$

This lemma shows that the upper bound depends on the sequence of the one-step gradient of the surrogate objective function over $K$ episodes which can be shown that be further bounded by $\tilde{\mathcal{O}}(\sqrt{K})$. Then, we will introduce the following lemma to specify how the term $\sqrt{\sum_{k=1}^{K}\|\nabla_k - \nabla_{k-1}\|^2}$ can be bounded.

**Lemma 5.9.** *Let $\nabla_k = \nabla f_k(q_k)$ denote the subgradient of the surrogate objective function $f_k$ evaluated at $q_k$ (Eq. (19)). Under OMDPD, the cumulative variation of consecutive gradients is bounded as:*

$$\sqrt{\sum_{k=1}^{K}\|\nabla_k - \nabla_{k-1}\|_2^2} \leqslant \sqrt{\sum_{k=1}^{K}\|\tilde{r}_k - \tilde{r}_{k-1}\|^2}\,(\mathbf{i})$$

$$+ \sqrt{\sum_{k=1}^{K}\|\Phi'(\lambda_k)\tilde{d}_k - \Phi'(\lambda_{k-1})\tilde{d}_{k-1}\|^2}\,(\mathbf{ii})$$

$$+ \sqrt{2\sum_{k=1}^{K}\|\tilde{r}_k - \tilde{r}_{k-1}\|\|\Phi'(\lambda_k)\tilde{d}_k - \Phi'(\lambda_{k-1})\tilde{d}_{k-1}\|}\,(\mathbf{iii})$$

$$\leqslant \frac{\sqrt{6SAHK}(1 + \Phi'(\lambda_K))}{SAH}$$

This lemma first establishes that the aggregated variation of policy gradients under OMDPD can be bounded by $\tilde{\mathcal{O}}(SAH\sqrt{K}) + \Phi'(\lambda_K)SAH\sqrt{K}$. We also recall the foundational inequality: $\Phi(\lambda_K) + \alpha \sum_{k=1}^{K}(\tilde{r}_k^\top q^* - \tilde{r}_k^\top q_k) \leq \sum_{k=1}^{K}(f_k(q_k) - f_k(q^*))$. Here, for simplicity, we momentarily ignore the factor of $SAH$ to discuss how the $\mathcal{O}(1)$ result in Remark 5.2 can be achieved. First, choosing the function $\Phi(x) = \exp(\beta x) - 1$ ensures that $\Phi'(\lambda_K)$ can be combined with $\Phi(\lambda_K)$ in the foundational inequality. Using Lemma 5.9, we obtain $\Phi(\lambda_K) + \alpha \sum_{k=1}^{K}(\tilde{r}_k^\top q^* - \tilde{r}_k^\top q_k) \leq \sqrt{K} + \Phi'(\lambda_K)\sqrt{K}$. Then we rearrange to get $\alpha \sum_{k=1}^{K}(\tilde{r}_k^\top q^* - \tilde{r}_k^\top q_k) \leq \exp(\beta\lambda_K)(\beta\sqrt{K} - 1) + 1 + \sqrt{K}$. By choosing the $\beta$ from Theorem 5.1 so that $\beta\sqrt{K} - 1 \leq 0$, it follows that $\sum_{k=1}^{K}(\tilde{r}_k^\top q^* - \tilde{r}_k^\top q_k) \leq \sqrt{K}$, which has an $\mathcal{O}(\sqrt{K})$ bound. Now, when the reward is fixed, terms (i) and (iii) vanish (because $\tilde{r}_k = \tilde{r}_{k-1}$), which makes $\alpha \sum_{k=1}^{K}(\tilde{r}_k^\top q^* - \tilde{r}_k^\top q_k) \leq \exp(\beta\lambda_K)(\beta\sqrt{K} - 1) + 1$, so that the $\sqrt{K}$ factor disappears and a suitable $\beta$ yields an $\mathcal{O}(1)$ bound. The details for showing $\sum_{k=1}^{K}[\tilde{r}_k^\top q^* - \tilde{r}_k^\top q_k] \leq \mathcal{O}(1)$ are deferred to Appendix D.2. Consequently, using Lemmas 5.8 and 5.9, we can now move on to prove Theorem 5.1 in the following sections.

### 5.2. Proof of the Main Theorem

To prove the main theorem, we first know that the regret can be expressed as:

$$\text{Regret}(K) = \underbrace{\sum_{k=1}^{K}\left[V^{\pi_k}(\tilde{r}_k, \tilde{p}_k) - V^{\pi_k}(\bar{r}, p)\right]}_{\text{Estimation Error}}$$
$$+ \underbrace{\sum_{k=1}^{K}\left[V^{\pi^*}(\bar{r}, p) - V^{\pi_k}(\tilde{r}_k, \tilde{p}_k)\right]}_{\text{Optimization Error}}. \quad (24)$$

Similarly, the violation in stochastic setting can be formulated as:

$$\text{Violation}(K) = \underbrace{\sum_{k=1}^{K}\left[V^{\pi_k}(\bar{d}, p) - V^{\pi_k}(\tilde{d}_k, \tilde{p}_k)\right]^+}_{\text{Estimation Error}}$$
$$+ \underbrace{\sum_{k=1}^{K}\left[V^{\pi_k}(\tilde{d}_k, \tilde{p}_k)\right]^+}_{\text{Optimization Error}}. \quad (25)$$

In the constrained adversarial setting, we do not explicitly perform estimation on constraint $d$, so the only source of estimation error arises from the unknown transition kernel.

Overall, the decomposition operation separates the regret and violation into two distinct components: (i) *estimation error*, which arises due to inaccuracies in the estimated model

parameters, and (ii) *optimization error*, which is influenced by the online learning algorithm. In the following, we will analyze and bound each term individually.

To better illustrate the analysis of the main theorem, we provide a proof roadmap in Figure 1, which establishes the inequality $\Phi(\lambda_K) + \alpha \sum_{k=1}^{K}(\tilde{r}_k^\top q^* - \tilde{r}_k^\top q_k) \leq Regret_{alg}$, along with the resulting regret and violation bounds.

### 5.3. Upper Bound of Estimation Error

In the following lemma, we provide an upper bound on the estimation errors for both regret and violation.

**Lemma 5.10.** *Let $\tilde{p}_k$ denote the transition kernel in the candidate set $B_k$, and let $\tilde{r}_k$ and $\tilde{d}_k$ be the estimations used by OMDPD. Then, conditioned on the good event $G$, the estimation errors for the stochastic cost case can be bounded as follows:*

$$\sum_{k=1}^{K}[V^{\pi_k}(\tilde{r}_k, \tilde{p}_k) - V^{\pi_k}(\bar{r}, p)]$$
$$\leq \tilde{\mathcal{O}}(\sqrt{\mathcal{N}SAH^3K} + S^2AH^3),$$
$$\sum_{k=1}^{K}\left[V^{\pi_k}(\bar{d}, p) - V^{\pi_k}(\tilde{d}_k, \tilde{p}_k)\right]^+$$
$$\leq \tilde{\mathcal{O}}(\sqrt{\mathcal{N}SAH^3K} + S^2AH^3).$$

*The estimation error for the adversarial cost case can be bounded as follows:*

$$\sum_{k=1}^{K}[V^{\pi_k}(d_k, p) - V^{\pi_k}(d_k, \tilde{p}_k)]^+$$
$$\leq \tilde{\mathcal{O}}(\sqrt{\mathcal{N}SAH^3K} + S^2AH^3).$$

The above lemma bounds the error incurred by using the estimated transition kernels, rewards, and costs during the learning process, which improves upon Lemma 29 of (Efroni et al., 2020) by a factor of $\tilde{\mathcal{O}}(\sqrt{H})$. This improvement is achieved by leveraging a Bellman-type law of total variance to control the expected sum of value estimates for bounding the error from estimating the transition kernel (Azar et al., 2017; Chen & Luo, 2021). The detailed proof is deferred in Appendix C.2. Next, we will bound the optimization error.

### 5.4. Upper Bound of Optimization Error

**Regret Analysis.** We first focus on bounding the optimization error associated with regret. The following lemma establishes that the cumulative variation of gradients between consecutive episodes under OMDPD is bounded, enabling adaptive regret-violation guarantees.

To further relate the regret optimization error to Algorithm

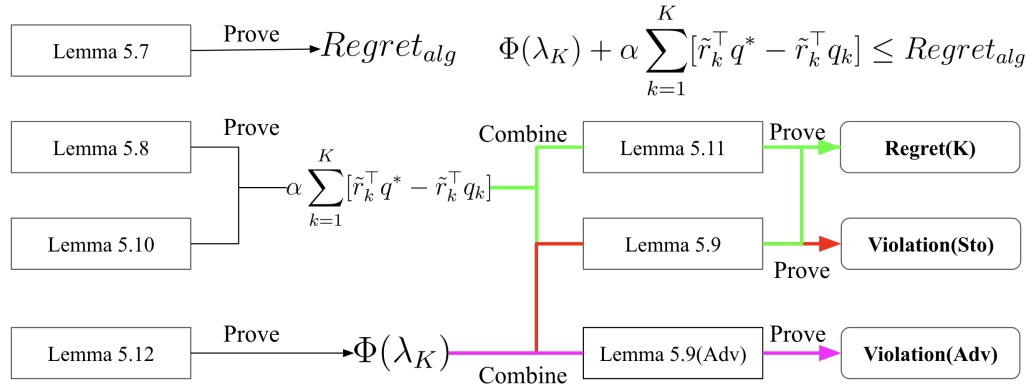

Figure 1. Proof Roadmap of the Theorem 5.1

1, consider:

$$\sum_{k=1}^{K}\left[V^{\pi^*}(\bar{r},p)-V^{\pi_k}(\tilde{r}_k,\tilde{p}_k)\right]=\sum_{k=1}^{K}\left[\mathbb{E}[\bar{r}^\top q^*]-\mathbb{E}[\tilde{r}_k^\top q_k]\right]$$

$$=\underbrace{\sum_{k=1}^{K}\left[\mathbb{E}[\bar{r}^\top q^*]-\mathbb{E}[\tilde{r}_k^\top q^*]\right]}_{\text{Term 1}}+\underbrace{\sum_{k=1}^{K}\left[\mathbb{E}[\tilde{r}_k^\top q^*]-\mathbb{E}[\tilde{r}_k^\top q_k]\right]}_{\text{Term 2}}.$$

(26)

Notably, *Term 2* corresponds to the violation-regret relationship in Eq. (23), providing a critical link to our theoretical analysis. Consequently, we will first derive an upper bound for *Term 2* in the regret decomposition.

**Lemma 5.11.** *Based on Lemma 5.8, 5.9, the following upper bound holds:*

$$\sum_{k=1}^{K}\left[\tilde{r}_k^\top q^*-\tilde{r}_k^\top q_k\right]$$
$$\leqslant\ 2(1+\sqrt{L_\delta})(SAH+4\sqrt{\mathcal{C}}\sqrt{6SAHK})$$

To complete the upper bound for the optimization error in regret, we now consider *Term 1*. The following lemma provides the required result:

**Lemma 5.12.** *Under the stochastic rewards setting, with probability at least 1-2δ, we have:*

$$\sum_{k=1}^{K}\left[\bar{r}^\top q^*-\tilde{r}_k^\top q^*\right]\leqslant SAH\sqrt{\frac{(K-1)}{2}\ln(\frac{2}{\delta})}+SAH$$

By combining *Term 1* and *Term 2*, we can obtain the bound of $\sum_{k=1}^{K}\left[V^{\pi^*}(\bar{r},p)-V^{\pi_k}(\tilde{r}_k,\tilde{p}_k)\right]$. Finally, by incorporating the estimation error bounds from Lemma 5.10, we can establish the complete regret bound, combining both the estimation and optimization errors. The detailed proofs of Lemmas 5.9, 5.11 and 5.12 can be found in Appendix C.3, C.4, C.5.

**Violation Analysis.** We now analyze the sublinear violation guarantee in stochastic and adversarial settings.

**Stochastic Setting.** As discussed earlier, in the stochastic setting, the overall violation can be decomposed into estimation and optimization parts, with Lemma 5.10 addressing the estimation error. We now turn our attention to bounding the optimization error. The following lemma provides the critical result needed for analyzing this optimization error.

**Lemma 5.13.** *Based on Lemma 5.8, 5.9. Then, the following upper bound holds:*

$$\sum_{k=1}^{K}\left[\mathbf{d}_k^\top q_k\right]^+\leqslant 16(1+\sqrt{L_\delta})\sqrt{\mathcal{C}}\sqrt{6SAHK}\ln(A)$$

*where* $A=\left(K+8\sqrt{\mathcal{C}}\left(\frac{\sqrt{6SAHK}}{SAH}\right)+2\right); \mathbf{d}_k=\tilde{d}_k$ *under stochastic setting and* $\mathbf{d}_k=d_k$ *in adversarial setting.*

Thus, by combining the estimation bound from Lemma 5.10 with the violation analysis from Lemma 5.13, we have derived the complete violation bound for the stochastic setting.

**Adversarial Setting.** As defined in Eq.(6), the estimation error in the adversarial setting differs slightly from that in the stochastic setting, since there is no estimation error associated with the constraints and the only source of estimation error arises from the transition kernel. Consequently, based on Lemma 5.10 under the adversarial case and Lemma 5.13, we obtain the complete violation bound for the adversarial setting. A detailed proof of Lemma 5.13 can be found in Appendix C.6.

## 6. Simulation

We evaluate our algorithm in a synthetic and finite-horizon CMDP environment constructed to assess performance under both stochastic and adversarial cost settings. The CMDP consists of a state space $\mathcal{S}=\{0,1,2,3,4\}$ with five discrete states and an action space $\mathcal{A}=\{0,1,2\}$ with three available actions. The decision process unfolds over a fixed horizon of

$H = 5$ steps. At each time step, the agent receives a reward $r \in [0, 1]^{H \times S \times A}$ sampled uniformly from the unit interval. In the stochastic setting, the cost $c \in [-1, 1]^{H \times S \times A}$ is also drawn uniformly and held fixed across episodes. In contrast, the adversarial setting introduces a discrete cost perturbation mechanism: in each episode, the cost is independently sampled from a finite set $\{-1.0, -0.6, -0.2, 0.0, 0.2, 0.6, 1.0\}$, simulating abrupt shifts in constraint feedback. The transition dynamics are time-dependent, where each transition distribution $P_{h,s,a}$ is independently sampled from a Dirichlet distribution with a concentration parameter $\alpha = 0.5$. A smaller concentration parameter like 0.5 encourages sparsity in the resulting probability vectors, meaning that the sampled distributions are likely to concentrate mass on a small subset of next states. This induces partially deterministic behavior while still preserving stochasticity across transitions. The initial state is sampled uniformly, ensuring that each trajectory starts from a randomly selected state. Throughout all experiments, the cumulative cost constraint threshold is 0. This controlled CMDP environment enables us to evaluate our algorithm under both stochastic and adversarial constraint settings. We plot the cumulative constraint violation across learning episodes($K = 3000$), where both the stochastic and adversarial curves clearly demonstrate the algorithm's ability to ensure sublinear violation growth. In particular, the observed trend aligns with the theoretical $\mathcal{O}(\sqrt{K})$, highlighting the algorithm's robustness in maintaining feasibility over time.

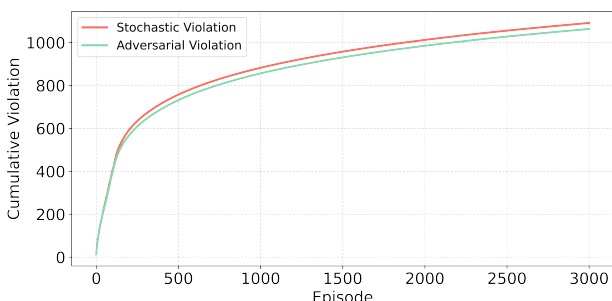

*Figure 2.* Cumulative Violation over Learning Episodes

## 7. Conclusion

In this work, we addressed the challenge of online safe reinforcement learning in dynamic environments with adversarial constraints by proposing the **O**ptimistic **M**irror **D**escent **P**rimal-**D**ual (OMDPD) algorithm. Our approach is the first to provide optimal guarantees in terms of both regret and strong constraint violation under anytime adversarial cost functions, without requiring Slater's condition or the existence of a strictly known safe policy. OMDPD achieves regret and violation bounds of $\tilde{\mathcal{O}}(\sqrt{K})$, which are optimal with respect to the number of learning episodes $K$. We also demonstrated that access to accurate estimates of rewards and transitions can further improve these per-

formance guarantees. Our work advances the theoretical understanding of CMDPs and provides a robust solution for safe decision-making in adversarial and non-stationary environments. Future research directions include extending our framework to multi-agent settings and investigating scenarios with partial observability.

## Impact Statement

This paper presents work whose goal is to advance the field of Machine Learning. There are many potential societal consequences of our work, none of which we feel must be specifically highlighted here.

## Acknowledgment

This work was supported by the National Research Foundation of Korea (NRF) grants (No. RS-2024-00350703 and No. RS-2024-00410082).

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

## A. Missing Related Work

**Online Constrained Optimization.** Recently, (Sinha & Vaze, 2024) achieved sublinear regret in an adversarial violation setting. However, their approach relies on Online Gradient Descent and thus cannot attain a tighter bound even when the reward is fixed. Meanwhile, Lekeufack & Jordan (2024) proposed an algorithm based on optimistic online mirror descent, attaining comparable regret and violation bounds to ours.

## B. Optimistic estimates Related Lemmas

### B.1. Proof of Lemma 5.6

*Lemma* 5.6. With probability at least $1 - \delta$, $\Pr[G] \geqslant 1 - \delta$, where $G$ is the good event defined in Eq. (27) for stochastic constraint setting and Eq.(28) for adversarial setting, where $\delta \in (0, 1)$.

*Proof:*

Define the following failure events representing the set in which the transitions and observations are far from our current optimistic-estimation:

$$F_k^p = \left\{ \exists s, a, s', h : \left| p_h(s'|s, a) - \hat{p}_h^{k-1}(s'|s, a) \right| > \beta_{k,h}^p(s, a, s') \right\}$$

$$F^N = \left\{ \sum_{k=1}^K \sum_{(s,a,h)} \frac{q_h^{\pi_k}(s, a|p)}{n_h^{k-1}(s, a) \vee 1} > 4HSA + 2HSA \ln K + 4 \ln \frac{2HK}{\delta'}, \right.$$

$$\left. \sum_{k=1}^K \sum_{(s,a,h)} \frac{q_h^{\pi_k}(s, a|p)}{\sqrt{n_h^{k-1}(s, a) \vee 1}} > 6HSA + 2H\sqrt{SAK} + 2HSA \ln K + 5 \ln \frac{2HK}{\delta'} \right\}$$

$$F_k^r = \left\{ \exists s, a, h : \left| \bar{r}_h(s, a) - \hat{r}_h^k(s, a) \right| > \beta_{k,h}^r(s, a) \right\}$$

$$F_k^d = \left\{ \exists s, a, h : \left| \bar{d}_h(s, a) - \hat{d}_h^k(s, a) \right| > \beta_{k,h}^d(s, a) \right\} \text{ (Stochastic setting)}$$

we define $\beta_{k,h}^p$ as

$$\beta_{k,h}^p(s, a, s') := 2\sqrt{\frac{\hat{p}_h^{k-1}(s'|s, a)(1 - \hat{p}_h^{k-1}(s'|s, a))L_\delta^p}{n_h^{k-1}(s, a) \vee 1}} + \frac{\frac{14}{3}L_\delta^p}{n_h^{k-1}(s, a) \vee 1}$$

$$\beta_{k,h}^r(s, a) := \beta_{k,h}^d(s, a) := \sqrt{\frac{L_\delta}{n_h^{k-1}(s, a) \vee 1}}$$

where we set $L_\delta = \ln(\frac{12SAHK}{\delta}), L_\delta^p = \ln(\frac{6SAHK}{\delta})$ and $\delta' = \frac{\delta}{3}$. Then set $F^p := \bigcup_{k=1}^K F_k^p, F^r = \bigcup_{k=1}^K F_k^r, F^d = \bigcup_{k=1}^K F_k^d$. Hence the **Good event** is denoted as:

$$G := \overline{\left( F^N \bigcup F^p \bigcup F^r \bigcup F^d \right)} \tag{27}$$

And because we did not estimate constraint $d$ in the adversarial setting, the **Good event** is defined as:

$$G := \overline{\left( F^N \bigcup F^p \bigcup F^r \right)} \tag{28}$$

Finally, it is easy to show that the good event $G$ happens with probability at least $1 - \delta$. Specifically, $\Pr[F^p \cup F^r \cup F^d] \leqslant \frac{2}{3}\delta$, where the detailed proof can be found in (Efroni et al., 2020)(Appendix A.1). Furthermore, by Lemma E.2, $\Pr[F^N] \leqslant \delta' = \frac{1}{3}\delta$. Then we can prove that $\Pr[G] \geqslant 1 - \delta$ by union bound.

### B.2. Proof of Lemma 5.7

*Lemma* 5.7. Conditioning on the **Good event** $G$, the optimal policy $\pi^*$ induced by the occupancy measure $q^{\pi^*}$ under the optimal policy for solving the CMDP problem (3) is a feasible solution policy for any episode $k \in [K]$ such that:

$$\pi^* \in \left\{ \pi \in \Delta_{\mathcal{A}}^{\mathcal{S}} : \bar{d}_k^\top q^\pi(p') \leqslant 0, p' \in B_k \right\}$$

*Proof:* For the stochastic cost case, the good event $G$ implies that $|\bar{d}_h(s,a) - \hat{d}_{k,h}(s,a)| \leqslant \beta_{k,h}^d(s,a)$ for all $(s,a,h,k) \in \mathcal{S} \times \mathcal{A} \times \mathcal{S} \times [K]$. Due to the definition of the optimistic cost $\tilde{d}_k$, we have $\tilde{d}_{k,h}(s,a) \leqslant \bar{d}_h(s,a)$. Then we have $\tilde{d}_k^\top q^* \leqslant \bar{d}^\top q^*$. Since $\pi^*$ is a feasible solution of (3), we have $\tilde{d}_k^\top q^* \leqslant \bar{d}^\top q^* \leqslant 0$. For the adversarial cost case, we take $\tilde{d}_k = d_k$. Furthermore, conditioned on the good event $G$, we know that the true transition kernel $p \in B_k$. Therefore, $q^*$ satisfies that $q^* \in \left\{ q : \tilde{d}_k^\top q(p') \leqslant 0, p' \in B_k \right\}$.

## C. Key Lemmas proofs for Theorem 5.1

### C.1. Proof of Lemma 5.8

*Lemma* 5.8. Let $\mathcal{C} = \sup_{q_1,q_2 \in Q} \mathcal{D}(q_1,q_2)$, $\nabla_k = \nabla(f_k(q_k))$, $\nabla_{k-1} = \nabla(f_{k-1}(q_{k-1}))$, and define the learning rate as:
$\eta_k = \sqrt{\mathcal{C}} \min \left\{ \frac{1}{\sqrt{\sum_{i=1}^{k-1} \|\nabla_i - \nabla_{i-1}\|_2^2} + \sqrt{\sum_{i=1}^{k-2} \|\nabla_i - \nabla_{i-1}\|_2^2}}, 1 \right\}$. Then, the regret is bounded as:

$$Regret_{alg} \leqslant 3.5\sqrt{\mathcal{C}} \left( \sqrt{\sum_{k=1}^{K} \|\nabla_k - \nabla_{k-1}\|_2^2 + 1} \right)$$

*Proof:* Updating rule we use for Optimistic OMD in Algorithm 1:

$$f_k(q) = \alpha(-\tilde{r}_k^\top q + \Phi'(\lambda_k)[\tilde{d}_k^\top q]^+) - \frac{1}{2}\|q - q_k\|^2$$

where we know $f_k(q)$ is 1-strong convex. By apply Optimistic OMD and convexity of $f_k(q)$, we have:

$$(f_k(q_k) - f_k(q^*)) \leqslant \langle \nabla(f_k(q_k)), q_k - q^* \rangle$$

For easy notation, we denote $\nabla(f_k(q_k)) = \nabla_k, \nabla(f_{k-1}(q_{k-1})) = \nabla_{k-1}$ Then, we can arrange for the following equal transformation:

$$\langle q_k - q^*, \nabla_k \rangle = \overbrace{\langle q_k - \hat{q}_k, \nabla_k - \nabla_{k-1} \rangle}^{\text{term 1}} + \overbrace{\langle q_k - \hat{q}_k, \nabla_{k-1} \rangle}^{\text{term 2}} + \overbrace{\langle \hat{q}_k - q^*, \nabla_k \rangle}^{\text{term 3}} \tag{29}$$

We can directly have upper bound for term 1:

$$\langle q_k - \hat{q}_k, \nabla_k - \nabla_{k-1} \rangle \leqslant \|q_k - \hat{q}_k\|_2 \|\nabla_k - \nabla_{k-1}\|_2$$

And any update of the form $a^* = \arg\min_{a \in A} \eta\langle a, x \rangle + \mathcal{D}(a, c)$ satisfies for any $d \in A$:

$$\langle a^* - d, x \rangle \leqslant \frac{1}{\eta} \left( \mathcal{D}(d, c) - \mathcal{D}(d, a^*) - \mathcal{D}(a^*, c) \right)$$

In our form, replace $a^* = q_k, d = \hat{q}_k, c = \hat{q}_{k-1}, x = \nabla_{k-1}, \eta = \eta_k$, we have upper bound for term 2:

$$\langle q_k - \hat{q}_k, \nabla_{k-1} \rangle \leqslant \frac{1}{\eta_k} \left( \mathcal{D}(\hat{q}_k, \hat{q}_{k-1}) - \mathcal{D}(\hat{q}_k, q_k) - \mathcal{D}(q_k, \hat{q}_{k-1}) \right) \tag{30}$$

Replace $a^* = \hat{q}_k, d = q^*, c = \hat{q}_{k-1}, x = \nabla_k, \eta = \eta_k$, we have upper bound for term 3:

$$\langle \hat{q}_k - q^*, \nabla_k \rangle \leqslant \frac{1}{\eta_k} \left( \mathcal{D}(q^*, \hat{q}_{k-1}) - \mathcal{D}(q^*, \hat{q}_k) - \mathcal{D}(\hat{q}_k, \hat{q}_{k-1}) \right) \tag{31}$$

Combine their upper bound together we have:

$$\langle q_k - q^*, \nabla_k \rangle \leqslant \|q_k - \hat{q}_k\|_2 \|\nabla_k - \nabla_{k-1}\|_2 + \frac{1}{\eta_k} \left[ \mathcal{D}(\hat{q}_k, \hat{q}_{k-1}) - \mathcal{D}(\hat{q}_k, q_k) - \mathcal{D}(q_k, \hat{q}_{k-1}) \right]$$

$$+ \frac{1}{\eta_k} \left[ \mathcal{D}(q^*, \hat{q}_{k-1}) - \mathcal{D}(q^*, \hat{q}_k) - \mathcal{D}(\hat{q}_k, \hat{q}_{k-1}) \right] \tag{32}$$

$$\leqslant \|q_k - \hat{q}_k\|_2 \|\nabla_k - \nabla_{k-1}\|_2 + \frac{1}{\eta_k} \left[ \mathcal{D}(q^*, \hat{q}_{k-1}) - \mathcal{D}(q^*, \hat{q}_k) - \mathcal{D}(\hat{q}_k, q_k) - \mathcal{D}(q_k, \hat{q}_{k-1}) \right]$$

Because we set $U$ is a 1-strongly convex function, we have $\mathcal{D}(q_1, q_2) \geqslant \frac{1}{2}\|q_1 - q_2\|_2^2$; then:

$$\langle q_k - q^*, \nabla_k \rangle \leqslant \|q_k - \hat{q}_k\|_2 \|\nabla_k - \nabla_{k-1}\|_2 + \frac{1}{\eta_k} \left[ \mathcal{D}(q^*, \hat{q}_{k-1}) - \mathcal{D}(q^*, \hat{q}_k) - \frac{1}{2}\|\hat{q}_k - q_k\|_2^2 - \frac{1}{2}\|q_k - \hat{q}_{k-1}\|_2^2 \right]$$

Then, sum it:

$$\sum_{k=1}^{K} \langle q_k - q^*, \nabla_k \rangle \leqslant \sum_{k=1}^{K} \|q_k - \hat{q}_k\|_2 \|\nabla_k - \nabla_{k-1}\|_2 + \frac{1}{\eta_1} \mathcal{D}(q^*, \hat{q}_0) + \sum_{k=2}^{K} \mathcal{D}(q^* - \hat{q}_{k-1})(\frac{1}{\eta_k} - \frac{1}{\eta_{k-1}})$$
$$- \sum_{k=1}^{K} \frac{1}{2\eta_k} (\|q_k - \hat{q}_k\|_2^2 + \|q_k - \hat{q}_{k-1}\|_2^2)$$

We define $\mathcal{C} = \sup_{q_1, q_2 \in \mathcal{Q}} \mathcal{D}(q_1, q_2)$, then we have:

$$\sum_{k=1}^{K} \langle q_k - q^*, \nabla_k \rangle \leqslant \overbrace{(\frac{1}{\eta_1} + \frac{1}{\eta_K})\mathcal{C}}^{\text{term (a)}} + \overbrace{\sum_{k=1}^{K} \|q_k - \hat{q}_k\|_2 \|\nabla_k - \nabla_{k-1}\|_2}^{\text{term (b)}} - \overbrace{\sum_{k=1}^{K} \frac{1}{2\eta_k} (\|q_k - \hat{q}_k\|_2^2 + \|q_k - \hat{q}_{k-1}\|_2^2)}^{\text{term (c)}}$$

Besides, we will use the following fact:

$$\|q_k - \hat{q}_k\|_2 \|\nabla_k - \nabla_{k-1}\|_2 = \inf_{\rho > 0} \left\{ \frac{\rho}{2} \|\nabla_k - \nabla_{k-1}\|_2^2 + \frac{1}{2\rho} \|q_k - \hat{q}_k\|_2^2 \right\}$$

And by setting $\rho = \eta_{k+1}$, we have upper bound for term (b):

$$\|q_k - \hat{q}_k\|_2 \|\nabla_k - \nabla_{k-1}\|_2 \leqslant \frac{\eta_{k+1}}{2} \|\nabla_k - \nabla_{k-1}\|_2^2 + \frac{1}{2\eta_{k+1}} \|q_k - \hat{q}_k\|_2^2$$

Then, consider learning rate $\eta_k$ in the following condition:

$$\eta_k = \sqrt{\mathcal{C}} \min \left\{ \frac{1}{\sqrt{\sum_{i=1}^{k-1} \|\nabla_i - \nabla_{i-1}\|_2^2} + \sqrt{\sum_{i=1}^{k-2} \|\nabla_i - \nabla_{i-1}\|_2^2}}, 1 \right\}$$

$$\eta_k \geqslant \sqrt{\mathcal{C}} \min \left\{ \frac{1}{2\sqrt{\sum_{i=1}^{k-1} \|\nabla_i - \nabla_{i-1}\|_2^2}}, 1 \right\}$$

$$\frac{1}{\eta_k} \leqslant \frac{1}{\sqrt{\mathcal{C}}} \max \left\{ 2\sqrt{\sum_{i=1}^{k-1} \|\nabla_i - \nabla_{i-1}\|_2^2}, 1 \right\}$$

Then for term (a), the upper bound is $(\frac{1}{\eta_1} + \frac{1}{\eta_K})\sqrt{\mathcal{C}} \leqslant \sqrt{\mathcal{C}}(2\sqrt{\sum_{k=1}^{K-1} \|\nabla_k - \nabla_{k-1}\|_2^2} + 2)$. Now, we get:

$$\sum_{k=1}^{K} \langle q_k - q^*, \nabla_k \rangle \leqslant \sqrt{\mathcal{C}} \left( 2\sqrt{\sum_{k=1}^{K} \|\nabla_k - \nabla_{k-1}\|_2^2} + 2 \right) + \sum_{k=1}^{K} \frac{\eta_{k+1}}{2} \|\nabla_k - \nabla_{k-1}\|_2^2$$
$$+ \sum_{k=1}^{K} \frac{1}{2\eta_{k+1}} \|q_k - \hat{q}_k\|_2^2 - \sum_{k=1}^{K} \frac{1}{2\eta_k} \|q_k - \hat{q}_k\|_2^2 - \sum_{k=1}^{K} \frac{1}{2\eta_k} \|q_k - \hat{q}_{k-1}\|_2^2$$
$$\leqslant \sqrt{\mathcal{C}} \left( 2\sqrt{\sum_{k=1}^{K} \|\nabla_k - \nabla_{k-1}\|_2^2} + 2 \right) + \sum_{k=1}^{K} \frac{\eta_{k+1}}{2} \|\nabla_k - \nabla_{k-1}\|_2^2$$
$$+ \sum_{k=1}^{K} \frac{1}{2\eta_{k+1}} \|q_k - \hat{q}_k\|_2^2 - \sum_{k=1}^{K} \frac{1}{2\eta_k} \|q_k - \hat{q}_k\|_2^2$$

where the second inequality arrived by dropping positive term $\|q_k - \hat{q}_{k-1}\|_2^2$. Now, we first deal with the last two terms:

$$\sum_{k=1}^{K} \frac{1}{2\eta_{k+1}} \|q_k - \hat{q}_k\|_2^2 - \sum_{k=1}^{K} \frac{1}{2\eta_k} \|q_k - \hat{q}_k\|_2^2 \leqslant \frac{\mathcal{C}}{2} \sum_{k=1}^{K} (\frac{1}{\eta_{k+1}} - \frac{1}{\eta_k}) \leqslant \frac{\mathcal{C}}{2\eta_{K+1}}$$

Now, we get:

$$\sum_{k=1}^{K} \langle q_k - q^*, \nabla_k \rangle \leqslant \sqrt{\mathcal{C}} \left( 2\sqrt{\sum_{k=1}^{K} \|\nabla_k - \nabla_{k-1}\|_2^2} + 2 \right) + \sum_{k=1}^{K} \frac{\eta_{k+1}}{2} \|\nabla_k - \nabla_{k-1}\|_2^2 + \frac{\mathcal{C}}{2\eta_{K+1}}$$

$$\leqslant 3\sqrt{\mathcal{C}} \left( \sqrt{\sum_{k=1}^{K} \|\nabla_k - \nabla_{k-1}\|_2^2} + 1 \right) + \sum_{k=1}^{K} \frac{\eta_{k+1}}{2} \|\nabla_k - \nabla_{k-1}\|_2^2$$

And notice $\eta_{k+1}$:

$$\eta_{k+1} = \sqrt{\mathcal{C}} \min \left\{ \frac{1}{\sqrt{\sum_{i=1}^{k} \|\nabla_i - \nabla_{i-1}\|_2^2} + \sqrt{\sum_{i=1}^{k-1} \|\nabla_i - \nabla_{i-1}\|_2^2}}, 1 \right\}$$

$$= \sqrt{\mathcal{C}} \min \left\{ \frac{\sqrt{\sum_{i=1}^{k} \|\nabla_i - \nabla_{i-1}\|_2^2} - \sqrt{\sum_{i=1}^{k-1} \|\nabla_i - \nabla_{i-1}\|_2^2}}{\|\nabla_k - \nabla_{k-1}\|_2^2}, 1 \right\}$$

Thus, we get:

$$\sum_{k=1}^{K} \frac{\eta_{k+1}}{2} \|\nabla_k - \nabla_{k-1}\|_2^2 \leqslant \frac{\sqrt{\mathcal{C}}}{2} \sum_{k=1}^{K} \left( \sqrt{\sum_{i=1}^{k} \|\nabla_i - \nabla_{i-1}\|_2^2} - \sqrt{\sum_{i=1}^{k-1} \|\nabla_i - \nabla_{i-1}\|_2^2} \right)$$

$$\leqslant \frac{\sqrt{\mathcal{C}}}{2} \sqrt{\sum_{k=1}^{K} \|\nabla_k - \nabla_{k-1}\|_2^2}$$

Next, we can have the upper bound:

$$\sum_{k=1}^{K} \langle q_k - q^*, \nabla_k \rangle \leqslant 3\sqrt{\mathcal{C}} \left( \sqrt{\sum_{k=1}^{K} \|\nabla_k - \nabla_{k-1}\|_2^2} + 1 \right) + \sum_{k=1}^{K} \frac{\eta_{k+1}}{2} \|\nabla_k - \nabla_{k-1}\|_2^2$$

$$\leqslant 3.5\sqrt{\mathcal{C}} \left( \sqrt{\sum_{k=1}^{K} \|\nabla_k - \nabla_{k-1}\|_2^2} + 1 \right)$$

Finally, we get:

$$\sum_{k=1}^{K} (f_k(q_k) - f_k(q^*)) \leqslant \sum_{k=1}^{K} \langle \nabla(f_k(q_k)), q_k - q^* \rangle = \sum_{k=1}^{K} \langle q_k - q^*, \nabla_k \rangle$$

$$\leqslant 3.5\sqrt{\mathcal{C}} \left( \sqrt{\sum_{k=1}^{K} \|\nabla_k - \nabla_{k-1}\|_2^2} + 1 \right)$$

### C.2. Proof of Lemma 5.10

*Lemma* 5.10. Let $\tilde{p}_k$ denote the transition kernel in the candidate set $B_k$, and let $\tilde{r}_k$ and $\tilde{d}_k$ be the estimations used by OMDPD. Then, conditioned on the good event $G$, the estimation errors for the stochastic cost case can be bounded as follows:

$$\sum_{k=1}^{K} \left[ V^{\pi_k}(\tilde{r}_k, \tilde{p}_k) - V^{\pi_k}(\bar{r}, p) \right] \leqslant \tilde{\mathcal{O}}(\sqrt{\mathcal{N}SAH^3K} + S^2AH^3),$$

$$\sum_{k=1}^{K} \left[ V^{\pi_k}(\bar{d}, p) - V^{\pi_k}(\tilde{d}_k, \tilde{p}_k) \right]^+ \leqslant \tilde{\mathcal{O}}(\sqrt{\mathcal{N}SAH^3K} + S^2AH^3). \tag{33}$$

The estimation error for the adversarial cost case can be bounded as follows:

$$\sum_{k=1}^{K} \left[ V^{\pi_k}(d_k, p) - V^{\pi_k}(d_k, \tilde{p}_k) \right]^+ \leqslant \tilde{\mathcal{O}}(\sqrt{\mathcal{N}SAH^3K} + S^2AH^3). \tag{34}$$

*Proof:* To prove (33), it is sufficient to show that

$$\sum_{k=1}^{K} \left| V^{\pi_k}(\tilde{\ell}_k, \tilde{p}_k) - V^{\pi_k}(\bar{\ell}, p) \right| \leq \tilde{\mathcal{O}} \left( \sqrt{\mathcal{N}SAH^3K} + S^2AH^3 \right)$$

for $\ell = r, d$. The right-hand side of the above inequality can be decomposed as

$$\sum_{k=1}^{K} \left| V^{\pi_k}(\tilde{\ell}_k, \tilde{p}_k) - V^{\pi_k}(\bar{\ell}, p) \right| \leq \underbrace{\sum_{k=1}^{K} \left| V^{\pi_k}(\tilde{\ell}_k, \tilde{p}_k) - V^{\pi_k}(\tilde{\ell}_k, p) \right|}_{\text{Term 1}} + \underbrace{\sum_{k=1}^{K} \left| V^{\pi_k}(\tilde{\ell}_k, p) - V^{\pi_k}(\bar{\ell}, p) \right|}_{\text{Term 2}}.$$

Note that $\beta_{k,h}^{\ell}(s,a) = \sqrt{L_\delta/(n_h^{k-1}(s,a) \vee 1)} \leq \sqrt{L_\delta}$. Then the estimated function $\tilde{\ell}_k$ satisfies $\tilde{\ell}_{k,h}(s,a) \in [-1 - \sqrt{L_\delta}, 1 + \sqrt{L_\delta}]$ for all $(s,a,h,k) \in \mathcal{S} \times \mathcal{A} \times [H] \times [K]$. By Lemma E.6 with $C = 1 + \sqrt{L_\delta}$, Term 1 is bounded as

$$\text{Term 1} \leq \tilde{\mathcal{O}} \left( \sqrt{\mathcal{N}SAH^3K} + S^2AH^3 \right).$$

To bound Term 2, by Lemma E.1, we can write it as

$$\text{Term 2} = \sum_{k=1}^{K} \mathbb{E} \left[ \sum_{h=1}^{H} \left| \bar{\ell}_h(s_h, a_h) - \tilde{\ell}_{k,h}(s_h, a_h) \right| \mid s_1, \pi_k, p \right].$$

Furthermore, conditioned on the good event $G$, it follows that

$$\left| \bar{\ell}_h(s,a) - \tilde{\ell}_{k,h}(s,a) \right| \leq \left| \bar{\ell}_h(s,a) - \hat{\ell}_h^{k-1}(s,a) \right| + \left| \hat{\ell}_h^{k-1}(s,a) - \tilde{\ell}_{k,h}(s,a) \right| \leq 2\sqrt{\frac{L_\delta}{n_h^{k-1}(s,a) \vee 1}}.$$

Applying this, Term 2 can be bounded as

$$\text{Term 2} \leq \sum_{k=1}^{K} \mathbb{E} \left[ \sum_{h=1}^{H} 2\sqrt{\frac{L_\delta}{n_h^{k-1}(s,a) \vee 1}} \mid s_1, \pi_k, p \right] \leq \tilde{\mathcal{O}} \left( H\sqrt{SAK} + HSA \right)$$

where the last inequality follows from Lemma E.2. Finally, we have

$$\sum_{k=1}^{K} \left| V^{\pi_k}(\tilde{\ell}_k, \tilde{p}_k) - V^{\pi_k}(\bar{\ell}, p) \right| = \text{Term 1} + \text{Term 2} \leq \tilde{\mathcal{O}} \left( \sqrt{\mathcal{N}SAH^3K} + S^2AH^3 \right)$$

as required.

Next, (34) is a direct consequence of Lemma E.6 with $C = 1$. $\qquad\square$

### C.3. Proof of Lemma 5.9

*Lemma* 5.9. Let $\nabla_k = \nabla f_k(q_k)$ denote the subgradient of the potential function $f_k$ evaluated at $q_k$. Under OMDPD, the cumulative variation of consecutive gradients is bounded as:

$$\sqrt{\sum_{k=1}^{K} \|\nabla_k - \nabla_{k-1}\|_2^2} \leq \frac{\sqrt{6SAHK}(1 + \Phi'(\lambda_K))}{SAH}$$

*Proof:* In the algorithm, we have:

$$\lambda_{k+1} = \lambda_k + \alpha[\tilde{d}_{k+1}^\top q_{k+1}]^+, \quad \Phi(x) = \exp(\beta x) - 1, \quad f_k(q) = \alpha(-\tilde{r}_k^\top q + \Phi'(\lambda_k)[\tilde{d}_k^\top q]^+) - \frac{1}{2}\|q - q_k\|^2$$

Based on convexity, we have:

$$\Phi(\lambda_k) \leq \Phi(\lambda_{k-1}) + \Phi'(\lambda_k)(\lambda_k - \lambda_{k-1})$$
$$= \Phi(\lambda_{k-1}) + \Phi'(\lambda_k) \cdot \alpha[\tilde{d}_k^\top q_k]^+$$

Based on drift analysis, we have:

$$\Phi(\lambda_k) - \Phi(\lambda_{k-1}) \leqslant \Phi'(\lambda_k) \cdot \alpha([\tilde{d}_k^\top q_k]^+) \tag{35}$$

And we know:

$$f_k(q_k) = \alpha(-\tilde{r}_k^\top q_k + \Phi'(\lambda_k)[\tilde{d}_k^\top q_k]^+) - \frac{1}{2}\|q_k - q_k\|^2$$

$$f_k(q_k) + \alpha(\tilde{r}_k^\top q_k) = \Phi'(\lambda_k) \cdot \alpha([\tilde{d}_k^\top q_k]^+) \tag{36}$$

Combine (35) and (36) together:

$$\Phi(\lambda_k) - \Phi(\lambda_{k-1}) \leqslant f_k(q_k) + \alpha(\tilde{r}_k^\top q_k)$$

Also, we have:

$$f_k(q^*) = \alpha(-\tilde{r}_k^\top q^* + \Phi'(\lambda_k)[\tilde{d}_k^\top q^*]^+) - \frac{1}{2}\|q^* - q_k\|^2$$

$$= -\alpha \cdot \tilde{r}_k^\top q^* - \frac{1}{2}\|q^* - q_k\|^2$$

Combine the equation together, we have:

$$\Phi(\lambda_k) - \Phi(\lambda_{k-1}) \leqslant f_k(q_k) + \alpha(\tilde{r}_k^\top q_k)$$

$$\Phi(\lambda_k) - \Phi(\lambda_{k-1}) - \alpha(\tilde{r}_k^\top q_k) - \hat{f}_k(q^*) \leqslant f_k(q_k) - f_k(q^*)$$

$$\Phi(\lambda_k) - \Phi(\lambda_{k-1}) + \alpha(\tilde{r}_k^\top q^* - \tilde{r}_k^\top q_k) \leqslant f_k(q_k) - f_k(q^*)$$

Take the summation over $K$, we have:

$$\Phi(\lambda_K) + \alpha \sum_{k=1}^{K} (\tilde{r}_k^\top q^* - \tilde{r}_k^\top q_k) \leqslant \sum_{k=1}^{K} f_k(q_k) - f_k(q^*)$$

Based on Lemma 5.8, we have:

$$\sum_{k=1}^{K} f_k(q_k) - f_k(q^*) \leqslant 3.5\sqrt{\mathcal{C}}\left(\sqrt{\sum_{k=1}^{K} \|\nabla_k - \nabla_{k-1}\|_2^2 + 1}\right)$$

Now, we analyze the upper bound part. $\nabla_k = \nabla f_k(q_k), \nabla_{k-1} = \nabla f_{k-1}(q_{k-1})$

$$\nabla_k = \nabla f_k(q_k) = \alpha(-\tilde{r}_k + \Phi'(\lambda_k)\tilde{d}_k)$$

$$\nabla_{k-1} = \nabla f_{k-1}(q_{k-1}) = \alpha(-\tilde{r}_{k-1} + \Phi'(\lambda_{k-1})\tilde{d}_{k-1})$$

Thus:

$$\|\nabla_k - \nabla_{k-1}\|^2 = \|\nabla f_k(q_k) - \nabla f_{k-1}(q_{k-1})\|^2$$

$$= \alpha^2\| - (\tilde{r}_k - \tilde{r}_{k-1}) + \Phi'(\lambda_k)\tilde{d}_k - \Phi'(\lambda_{k-1})\tilde{d}_{k-1}\|^2$$

$$= \alpha^2\|\tilde{r}_k - \tilde{r}_{k-1}\|^2 + \alpha^2\|\Phi'(\lambda_k)\tilde{d}_k - \Phi'(\lambda_{k-1})\tilde{d}_{k-1}\|^2 + 2\alpha\|\tilde{r}_k - \tilde{r}_{k-1}\|\|\Phi'(\lambda_k)\tilde{d}_k - \Phi'(\lambda_{k-1})\tilde{d}_{k-1}\|$$

$$\|\Phi'(\lambda_k)\tilde{d}_k - \Phi'(\lambda_{k-1})\tilde{d}_{k-1}\|^2 = \|\Phi'(\lambda_k)\tilde{d}_k - \Phi'(\lambda_{k-1})\tilde{d}_k + \Phi'(\lambda_{k-1})\tilde{d}_k - \Phi'(\lambda_{k-1})\tilde{d}_{k-1}\|^2$$

$$= \|\tilde{d}_k(\Phi'(\lambda_k) - \Phi'(\lambda_{k-1})) + \Phi'(\lambda_{k-1})(\tilde{d}_k - \tilde{d}_{k-1})\|^2$$

$$\leqslant 2(1 + \sqrt{L_\delta})^2 SAH\|\Phi'(\lambda_k) - \Phi'(\lambda_{k-1})\|^2 + 2(1 + \sqrt{L_\delta})^2 SAH\|\Phi'(\lambda_{k-1})\|^2$$

First deal with $\|\Phi'(\lambda_{k-1})\|^2$, we define $\Phi(x) = \exp(\beta x) - 1$, so we have $\Phi'(\lambda_k) = \beta\exp(\beta\lambda_k) > 0, \forall k \in K$. Because $\exp(x)$ is increasing, we have $\Phi'(\lambda_1) \leqslant \Phi'(\lambda_2) \leqslant ... \leqslant \Phi'(\lambda_K)$. Thus, $\forall k \in K$ we obtain:

$$\|\Phi'(\lambda_{k-1})\|^2 \leqslant \|\Phi'(\lambda_K)\|^2$$

For $\|\Phi'(\lambda_k) - \Phi'(\lambda_{k-1})\|^2$, we use $(a - b)^2 \leqslant a^2 + b^2$ and have:

$$\|\Phi'(\lambda_k) - \Phi'(\lambda_{k-1})\|^2 \leqslant \|\Phi'(\lambda_k)\|^2 + \|\Phi'(\lambda_{k-1})\|^2$$

$$\leqslant \|\Phi'(\lambda_K)\|^2 + \|\Phi'(\lambda_K)\|^2 = 2\|\Phi'(\lambda_K)\|^2$$

Therefore, we have the upper bound:

$$\|\Phi'(\lambda_k)\tilde{d}_k - \Phi'(\lambda_{k-1})\tilde{d}_{k-1}\|^2 \leqslant 2(1+\sqrt{L_\delta})^2 SAH\|\Phi'(\lambda_k) - \Phi'(\lambda_{k-1})\|^2 + 2(1+\sqrt{L_\delta})^2 SAH\|\Phi'(\lambda_{k-1})\|^2$$
$$\leqslant 2(1+\sqrt{L_\delta})^2 SAH \cdot 2\|\Phi'(\lambda_K)\|^2 + 2(1+\sqrt{L_\delta})^2 SAH \cdot \|\Phi'(\lambda_K)\|^2$$
$$\leqslant 6(1+\sqrt{L_\delta})^2 SAH\|\Phi'(\lambda_K)\|^2$$

Therefore,

$$\sqrt{\sum_{k=1}^{K}\|\nabla_k - \nabla_{k-1}\|_2^2}$$

$$= \alpha\sqrt{\sum_{k=1}^{K}\|\tilde{r}_k - \tilde{r}_{k-1}\|^2 + \sum_{k=1}^{K}\|\Phi'(\lambda_k)\tilde{d}_k - \Phi'(\lambda_{k-1})\tilde{d}_{k-1}\|^2 + 2\sum_{k=1}^{K}\|\tilde{r}_k - \tilde{r}_{k-1}\|\|\Phi'(\lambda_k)\tilde{d}_k - \Phi'(\lambda_{k-1})\tilde{d}_{k-1}\|}$$

$$\leqslant \underbrace{\alpha\sqrt{\sum_{k=1}^{K}\|\tilde{r}_k - \tilde{r}_{k-1}\|^2}}_{\text{diff 1}} + \underbrace{\alpha\sqrt{\sum_{k=1}^{K}\|\Phi'(\lambda_k)\tilde{d}_k - \Phi'(\lambda_{k-1})\tilde{d}_{k-1}\|^2}}_{\text{diff 2}} + \underbrace{\alpha\sqrt{2\sum_{k=1}^{K}\|\tilde{r}_k - \tilde{r}_{k-1}\|\|\Phi'(\lambda_k)\tilde{d}_k - \Phi'(\lambda_{k-1})\tilde{d}_{k-1}\|}}_{\text{diff 3}}$$

For diff 1:

$$\alpha\sqrt{\sum_{k=1}^{K}\|\tilde{r}_k - \tilde{r}_{k-1}\|^2} \leqslant \alpha\sqrt{(1+\sqrt{L_\delta})^2 SAHK}$$

For diff 2:

$$\alpha\sqrt{\sum_{k=1}^{K}\|\Phi'(\lambda_k)\tilde{d}_k - \Phi'(\lambda_{k-1})\tilde{d}_{k-1}\|^2} \leqslant \alpha\sqrt{\sum_{k=1}^{K}6(1+\sqrt{L_\delta})^2 SAH\|\Phi'(\lambda_K)\|^2} = \alpha\Phi'(\lambda_K)\sqrt{6(1+\sqrt{L_\delta})^2 SAHK}$$

For diff 3:

$$\sqrt{2\sum_{k=1}^{K}\|\tilde{r}_k - \tilde{r}_{k-1}\|\|\Phi'(\lambda_k)\tilde{d}_k - \Phi'(\lambda_{k-1})\tilde{d}_{k-1}\|} \leqslant \sqrt{2\sum_{k=1}^{K}\sqrt{(1+\sqrt{L_\delta})SAH}\|\Phi'(\lambda_k)\tilde{d}_k - \Phi'(\lambda_{k-1})\tilde{d}_{k-1}\|}$$

$$\leqslant \sqrt{2\sum_{k=1}^{K}\sqrt{(1+\sqrt{L_\delta})SAH}\sqrt{6(1+\sqrt{L_\delta})^2 SAH}\Phi'(\lambda_K)}$$

$$= \sqrt{\sum_{k=1}^{K}SAH\sqrt{24(1+\sqrt{L_\delta})^3}\Phi'(\lambda_K)}$$

$$\leqslant \sqrt{\sum_{k=1}^{K}SAH\sqrt{36(1+\sqrt{L_\delta})^4}\Phi'(\lambda_K)}$$

$$= \sqrt{\sum_{k=1}^{K}6(1+\sqrt{L_\delta})^2 SAH\Phi'(\lambda_K)} = \sqrt{\Phi'(\lambda_K)}\sqrt{6(1+\sqrt{L_\delta})^2 SAHK}$$

$$\leqslant \left(\Phi'(\lambda_K) + 1\right)\sqrt{6(1+\sqrt{L_\delta})^2 SAHK}$$

where the last inequality holds for $\sqrt{a} \leqslant a + 1, \forall a > 0$. Therefore, we have the upper bound of $\sqrt{\sum_{k=1}^{K}\|\nabla_k - \nabla_{k-1}\|_2^2}$:

$$\sqrt{\sum_{k=1}^{K}\|\nabla_k - \nabla_{k-1}\|_2^2} \leqslant \underbrace{\alpha\sqrt{\sum_{k=1}^{K}\|\tilde{r}_k - \tilde{r}_{k-1}\|^2}}_{\text{diff 1}} + \underbrace{\alpha\sqrt{\sum_{k=1}^{K}\|\Phi'(\lambda_k)\tilde{d}_k - \Phi'(\lambda_{k-1})\tilde{d}_{k-1}\|^2}}_{\text{diff 2}}$$

$$+\alpha\underbrace{\sqrt{2\sum_{k=1}^{K}\|\tilde{r}_k-\tilde{r}_{k-1}\|\|\Phi'(\lambda_k)\tilde{d}_k-\Phi'(\lambda_{k-1})\tilde{d}_{k-1}\|}}_{\text{diff 3}}$$

$$\leqslant\underbrace{\alpha\sqrt{(1+\sqrt{L_\delta})^2SAHK}}_{\text{diff 1}}+\underbrace{\alpha\Phi'(\lambda_K)\sqrt{6(1+\sqrt{L_\delta})^2SAHK}}_{\text{diff 2}}$$

$$+\underbrace{\alpha\sqrt{6(1+\sqrt{L_\delta})^2SAHK}+\alpha\Phi'(\lambda_K)\sqrt{6(1+\sqrt{L_\delta})^2SAHK}}_{\text{diff 3}}$$

$$\leqslant 2\alpha\sqrt{6(1+\sqrt{L_\delta})^2SAHK}+2\alpha\Phi'(\lambda_K)\sqrt{6(1+\sqrt{L_\delta})^2SAHK}=2\alpha(1+\sqrt{L_\delta})\sqrt{6SAHK}\left(1+\Phi'(\lambda_K)\right)$$

$$=\frac{\sqrt{6SAHK}(1+\Phi'(\lambda_K))}{SAH}$$

where the last equality holds for choosing $\alpha=\frac{1}{2(1+\sqrt{L_\delta})SAH}$.

### C.4. Proof of Lemma 5.11

*Lemma* 5.11. Based on Lemma 5.8, 5.9, the following upper bound holds:

$$\sum_{k=1}^{K}\left[\tilde{r}_k^\top q^*-\tilde{r}_k^\top q_k\right]\leqslant 2(1+\sqrt{L_\delta})(SAH+4\sqrt{\mathcal{C}}\sqrt{6SAHK})$$

*Proof:* Based on Lemma 5.8, 5.9, we have the following relation:

$$\Phi(\lambda_K)+\alpha\sum_{k=1}^{K}(\tilde{r}_k^\top q^*-\tilde{r}_k^\top q_k)\leqslant\sum_{k=1}^{K}\hat{f}_k(q_k)-\hat{f}_k(q^*)\leqslant 3.5\sqrt{\mathcal{C}}\left(\sqrt{\sum_{k=1}^{K}\|\nabla_k-\nabla_{k-1}\|_2^2+1}\right)$$

$$\leqslant 3.5\sqrt{\mathcal{C}}\left(\sqrt{\sum_{k=1}^{K}\|\nabla_k-\nabla_{k-1}\|_2^2}+1\right)$$

$$\leqslant 3.5\sqrt{\mathcal{C}}\left(\frac{\sqrt{6SAHK}(1+\Phi'(\lambda_K))}{SAH}\right)$$

Now we can have:

$$\Phi(\lambda_K)+\alpha\sum_{k=1}^{K}[\tilde{r}_k^\top q^*-\tilde{r}_k^\top q_k]\leqslant 3.5\sqrt{\mathcal{C}}\left(\frac{\sqrt{6SAHK}(1+\Phi'(\lambda_K))}{SAH}\right)$$

$$\Phi(\lambda_K)+\alpha\sum_{k=1}^{K}[\tilde{r}_k^\top q^*-\tilde{r}_k^\top q_k]\leqslant 3.5\sqrt{\mathcal{C}}\left(\frac{\sqrt{6SAHK}(1+\Phi'(\lambda_K))}{SAH}\right)$$

$$\exp(\beta\lambda_K)-1+\alpha\sum_{k=1}^{K}[\tilde{r}_k^\top q^*-\tilde{r}_k^\top q_k]\leqslant 4\sqrt{\mathcal{C}}\left(\frac{\sqrt{6SAHK}}{SAH}\right)+4\sqrt{\mathcal{C}}\left(\frac{\beta\exp(\beta\lambda_K)\sqrt{6SAHK}}{SAH}\right)$$

$$\alpha\sum_{k=1}^{K}[\tilde{r}_k^\top q^*-\tilde{r}_k^\top q_k]\leqslant\exp(\beta\lambda_K)\left(\frac{\beta\cdot 4\sqrt{\mathcal{C}}\sqrt{6SAHK}}{SAH}-1\right)+4\sqrt{\mathcal{C}}\left(\frac{\sqrt{6SAHK}}{SAH}\right)+1$$

$$\sum_{k=1}^{K}[\tilde{r}_k^\top q^*-\tilde{r}_k^\top q_k]\leqslant\exp(\beta\lambda_K)\left(\frac{\beta\cdot 4\sqrt{\mathcal{C}}\sqrt{6SAHK}}{\alpha SAH}-\frac{1}{\alpha}\right)+4\sqrt{\mathcal{C}}\left(\frac{\sqrt{6SAHK}}{\alpha SAH}\right)+\frac{1}{\alpha}$$

$$=2(1+\sqrt{L_\delta})SAH+8(1+\sqrt{L_\delta})\sqrt{\mathcal{C}}\sqrt{6SAHK}$$

$$=2(1+\sqrt{L_\delta})(SAH+4\sqrt{\mathcal{C}}\sqrt{6SAHK})$$

where the last equality obtained by $\alpha=\frac{1}{2(1+\sqrt{L_\delta})SAH},\beta\leqslant\frac{SAH}{4\sqrt{\mathcal{C}}\sqrt{6SAHK}}$.

## C.5. Proof of Lemma 5.12

*Lemma* 5.12. Under the stochastic rewards setting, with probability at least $1$-$2\delta$, we have:

$$\sum_{k=1}^{K} \left[ \bar{r}^\top q^* - \tilde{r}_k^\top q^* \right] \leqslant SAH \sqrt{\frac{K}{2} \ln(\frac{2}{\delta})}$$

*Proof:* Based on the optimistic estimates, we know that: $\tilde{r}_k = \hat{r}_{k-1} + \beta_{k-1}^r(s, a)$. Then, we have the following relationship:

$$\bar{r}^\top q^* - \tilde{r}_k^\top q^* = \bar{r}^\top q^* - \hat{r}_{k-1}^\top q^* - \beta_{k-1}^r(s, a) q^*$$
$$\leqslant \bar{r}^\top q^* - \hat{r}_{k-1}^\top q^*$$

where the inequality holds for $\beta_{k-1}^r(s, a)$ and $q^*$ is non-negative. Thus, we have the following relationship easily:

$$\sum_{k=1}^{K} \left[ \bar{r}^\top q^* - \tilde{r}_k^\top q^* \right] \leqslant \sum_{k=1}^{K} \left[ \bar{r}^\top q^* - \hat{r}_{k-1}^\top q^* \right]$$

Based on norm property, and for each episode $k$, we have:

$$\left| \bar{r}^\top q^* - \hat{r}_{k-1}^\top q^* \right| \leqslant \| \bar{r} - \hat{r}_{k-1} \|_\infty \cdot \| q^* \|_1$$

and from the definition of reward $r$ and **Fact 5.4**, we know that $\| \bar{r} - \hat{r}_{k-1} \|_\infty \leqslant 1$ and $\| q^* \|_1 \leqslant SAH$. Thus:

$$\left| \bar{r}^\top q^* - \hat{r}_{k-1}^\top q^* \right| \leqslant SAH$$

If the objective costs are stochastic under Lemma 5.6, and by the Azuma-Hoeffding inequality we have:

$$\Pr \left[ \left| \sum_{k=1}^{K-1} \bar{r}^\top q^* - \sum_{k=1}^{K-1} \hat{r}_{k-1}^\top q^* \right| \geqslant M \right] \leqslant \delta = 2e^{\left( -\frac{2M^2}{(K-1)(SAH)^2} \right)}$$

Thus by setting $M$ as:

$$M = SAH \sqrt{\frac{(K-1)}{2} \ln(\frac{2}{\delta})}$$

we have when the objective costs are stochastic, with the probability at least $1 - 2\delta$:

$$\left| \sum_{k=1}^{K-1} \bar{r}^\top q^* - \sum_{k=1}^{K-1} \hat{r}_{k-1}^\top q^* \right| \leqslant SAH \sqrt{\frac{(K-1)}{2} \ln(\frac{2}{\delta})}$$

Then, by the absolute value property, we can obtain:

$$\sum_{k=1}^{K} \left[ \bar{r}^\top q^* - \tilde{r}_k^\top q^* \right] \leqslant \sum_{k=1}^{K-1} \bar{r}^\top q^* - \sum_{k=1}^{K-1} \hat{r}_{k-1}^\top q^* + \bar{r}^\top q^*$$
$$\leqslant \left| \sum_{k=1}^{K-1} \bar{r}^\top q^* - \sum_{k=1}^{K-1} \hat{r}_{k-1}^\top q^* \right| + \left| \bar{r}^\top q^* \right|$$
$$\leqslant SAH \sqrt{\frac{(K-1)}{2} \ln(\frac{2}{\delta})} + SAH$$

## C.6. Proof of Lemma 5.13

*Lemma* 5.13. Based on Lemma 5.8, 5.9. Then, the following upper bound holds:

$$\sum_{k=1}^{K} \left[ \mathbf{d}_k^\top q_k \right]^+ \leqslant 16(1 + \sqrt{L_\delta}) \sqrt{\mathcal{C}} \sqrt{6SAHK} \ln \left( K + 8\sqrt{\mathcal{C}} \left( \frac{\sqrt{6SAHK}}{SAH} \right) + 2 \right)$$

where $\mathbf{d}_k = \tilde{d}_k$ under stochastic setting and $\mathbf{d}_k = d_k$ in adversarial setting.

*Proof:* To begin with, we deal with the stochastic setting first, which is $\mathbf{d}_k = \tilde{d}_k$. Since we have the fact that min and max value for regret that: $-|\tilde{r}_k^\top q^* - \tilde{r}_k^\top q_k| \leqslant (\tilde{r}_k^\top q^* - \tilde{r}_k^\top q_k) \leqslant |\tilde{r}_k^\top q^* - \tilde{r}_k^\top q_k|$. And $|\tilde{r}_k^\top q^* - \tilde{r}_k^\top q_k| \leqslant \|\tilde{r}_k\|_2 \|q^* - q_k\|_2 = (1 + \sqrt{L_\delta})SAH$. Thus, we have:

$$\sum_{k=1}^{K} (\tilde{r}_k^\top q^* - \tilde{r}_k^\top q_k) \geqslant \sum_{k=1}^{K} -|\tilde{r}_k^\top q^* - \tilde{r}_k^\top q_k| = -(1 + \sqrt{L_\delta})SAHK$$

Therefore, we have:

$$\Phi(\lambda_K) + \alpha \sum_{k=1}^{K} [\tilde{r}_k^\top q^* - \tilde{r}_k^\top q_k] \leqslant 3.5\sqrt{\mathcal{C}} \left( \frac{\sqrt{6SAHK}(1 + \Phi'(\lambda_K))}{SAH} \right)$$

$$\Phi(\lambda_K) + \alpha(-(1 + \sqrt{L_\delta})SAHK) \leqslant 3.5\sqrt{\mathcal{C}} \left( \frac{\sqrt{6SAHK}(1 + \Phi'(\lambda_K))}{SAH} \right)$$

$$\exp(\beta\lambda_K) - 1 + \alpha(-(1 + \sqrt{L_\delta})SAHK) \leqslant 4\sqrt{\mathcal{C}} \left( \frac{\sqrt{6SAHK}}{SAH} \right) + 4\sqrt{\mathcal{C}} \left( \frac{\beta \exp(\beta\lambda_K)\sqrt{6SAHK}}{SAH} \right)$$

$$\exp(\beta\lambda_K) \left( 1 - \beta \cdot 4\sqrt{\mathcal{C}} \left( \frac{\sqrt{6SAHK}}{SAH} \right) \right) \leqslant \alpha((1 + \sqrt{L_\delta})SAHK) + 4\sqrt{\mathcal{C}} \left( \frac{\sqrt{6SAHK}}{SAH} \right) + 1$$

$$\exp(\beta\lambda_K) \leqslant \frac{\alpha((1 + \sqrt{L_\delta})SAHK) + 4\sqrt{\mathcal{C}} \left( \frac{\sqrt{6SAHK}}{SAH} \right) + 1}{\left( 1 - \beta \cdot 4\sqrt{\mathcal{C}} \left( \frac{\sqrt{6SAHK}}{SAH} \right) \right)}$$

where the last inequality holds for choosing $\beta$ such that $1 - \beta \cdot 4\sqrt{\mathcal{C}} \left( \frac{\sqrt{6SAHK}}{SAH} \right) > 0$:

$$\beta \cdot 4\sqrt{\mathcal{C}} \left( \frac{\sqrt{6SAHK}}{SAH} \right) < 1 \rightarrow \beta < \frac{SAH}{4\sqrt{\mathcal{C}}\sqrt{6SAHK}}$$

which the choosing of $\beta$ match when we prove the regret bound in that case we choose $\beta \leqslant \frac{SAH}{4\sqrt{\mathcal{C}}\sqrt{6SAHK}}$. Here, we let $\beta = \frac{SAH}{8\sqrt{\mathcal{C}}\sqrt{6SAHK}}$ and we have:

$$\exp(\beta\lambda_K) \leqslant \frac{\alpha((1 + \sqrt{L_\delta})SAHK) + 4\sqrt{\mathcal{C}} \left( \frac{\sqrt{6SAHK}}{SAH} \right) + 1}{\left( 1 - \beta \cdot 4\sqrt{\mathcal{C}} \left( \frac{\sqrt{6SAHK}}{SAH} \right) \right)}$$

$$= \frac{\alpha((1 + \sqrt{L_\delta})SAHK) + 4\sqrt{\mathcal{C}} \left( \frac{\sqrt{6SAHK}}{SAH} \right) + 1}{\frac{1}{2}}$$

$$= 2\alpha((1 + \sqrt{L_\delta})SAHK) + 8\sqrt{\mathcal{C}} \left( \frac{\sqrt{6SAHK}}{SAH} \right) + 2$$

$$= K + 8\sqrt{\mathcal{C}} \left( \frac{\sqrt{6SAHK}}{SAH} \right) + 2$$

where the last inequality holds for take $\alpha = \frac{1}{2(1+\sqrt{L_\delta})SAH}$. Then, recall the definition $\lambda_k = \lambda_{k-1} + \alpha[\tilde{d}_k^\top q_k]^+$, so $\lambda_K = \alpha \sum_{k=1}^{K} [\tilde{d}_k^\top q_k]^+$. Thus, take the log operation and we have:

$$\beta\lambda_K \leqslant \ln \left( K + 8\sqrt{\mathcal{C}} \left( \frac{\sqrt{6SAHK}}{SAH} \right) + 2 \right)$$

$$\lambda_K \leqslant \frac{1}{\beta} \ln \left( K + 8\sqrt{\mathcal{C}} \left( \frac{\sqrt{6SAHK}}{SAH} \right) + 2 \right)$$

$$\alpha \sum_{k=1}^{K} [\tilde{d}_k^\top q_k]^+ \leqslant \frac{1}{\beta} \ln \left( K + 8\sqrt{\mathcal{C}} \left( \frac{\sqrt{6SAHK}}{SAH} \right) + 2 \right)$$

$$\sum_{k=1}^{K} [\tilde{d}_k^\top q_k]^+ \leqslant \frac{1}{\alpha\beta} \ln\left( K + 8\sqrt{\mathcal{C}}\left(\frac{\sqrt{6SAHK}}{SAH}\right) + 2\right)$$

$$= 16(1+\sqrt{L_\delta})\sqrt{\mathcal{C}}\sqrt{6SAHK} \ln\left( K + 8\sqrt{\mathcal{C}}\left(\frac{\sqrt{6SAHK}}{SAH}\right) + 2\right)$$

The proof of $\mathbf{d}_k = d_k$ is almost same with the situation that $\mathbf{d}_k = \tilde{d}_k$. Hence, we can directly have:

$$\sum_{k=1}^{K} [d_k^\top q_k]^+ \leqslant 16(1+\sqrt{L_\delta})\sqrt{\mathcal{C}}\sqrt{6SAHK} \ln\left( K + 8\sqrt{\mathcal{C}}\left(\frac{\sqrt{6SAHK}}{SAH}\right) + 2\right)$$

## D. Main Theoretical Analysis

### D.1. Proof of Theorem 5.1

*Proof:* In this section, our proof is based on the roadmap described in Figure 3. The context is divided into Regret and Violation parts, respectively.

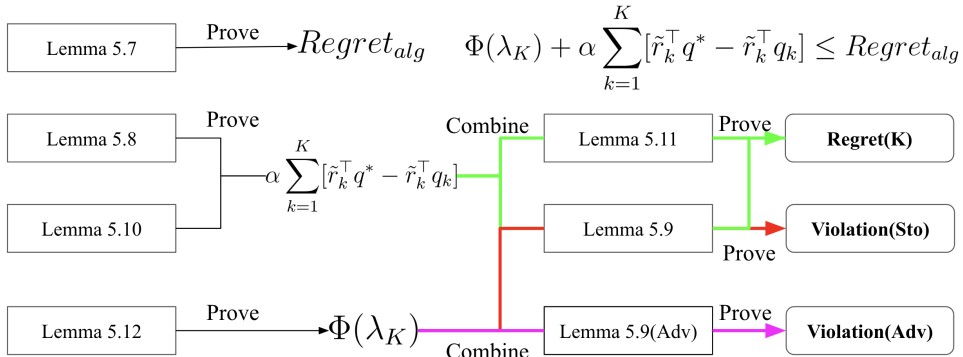

*Figure 3.* Proof Roadmap of Theorem 5.1

#### D.1.1. REGRET BOUND PROOF

Recall the definition of Regret:

$$\text{Regret}(K) = \sum_{k=1}^{K} \left[ V^{\pi^*}(\bar{r}, p) - V^{\pi_k}(\bar{r}, p) \right]$$

$$= \underbrace{\sum_{k=1}^{K} [V^{\pi_k}(\tilde{r}_k, \tilde{p}_k) - V^{\pi_k}(\bar{r}, p)]}_{\text{Estimation Error}} + \underbrace{\sum_{k=1}^{K} \left[ V^{\pi^*}(\bar{r}, p) - V^{\pi_k}(\tilde{r}_k, \tilde{p}_k) \right]}_{\text{Optimization Error}}$$

We can bound the "Estimation Error" term by using Lemma 5.10. Now, let's go through the details of the "Optimization Error" term. We will first decompose it as follows:

$$\sum_{k=1}^{K} \left[ V^{\pi^*}(\bar{r}, p) - V^{\pi_k}(\tilde{r}_k, \tilde{p}_k) \right] = \sum_{k=1}^{K} \left[ \mathbb{E}[\bar{r}^\top q^*] - \mathbb{E}[\tilde{r}_k^\top q_k] \right] = \underbrace{\sum_{k=1}^{K} \left[ \mathbb{E}[\bar{r}^\top q^*] - \mathbb{E}[\tilde{r}_k^\top q^*] \right]}_{\text{Term 1}} + \underbrace{\sum_{k=1}^{K} \left[ \mathbb{E}[\tilde{r}_k^\top q^*] - \mathbb{E}[\tilde{r}_k^\top q_k] \right]}_{\text{Term 2}}.$$

Thus, it's clear that we can use Lemma 5.11 and Lemma 5.12 to bound these two terms, respectively. Therefore, the Regret is bounded in the following inequality:

$$\text{Regret}(K) = \sum_{k=1}^{K} \left[ V^{\pi^*}(\bar{r}, p) - V^{\pi_k}(\bar{r}, p) \right] = \sum_{k=1}^{K} [V^{\pi_k}(\tilde{r}_k, \tilde{p}_k) - V^{\pi_k}(\bar{r}, p)] + \sum_{k=1}^{K} \left[ V^{\pi^*}(\bar{r}, p) - V^{\pi_k}(\tilde{r}_k, \tilde{p}_k) \right]$$

$$= \sum_{k=1}^{K} \underbrace{\left[ V^{\pi_k}(\tilde{r}_k, \tilde{p}_k) - V^{\pi_k}(\bar{r}, p) \right]}_{\text{Lemma 5.10}} + \sum_{k=1}^{K} \underbrace{\left[ \mathbb{E}[\bar{r}^\top q^*] - \mathbb{E}[\tilde{r}_k^\top q^*] \right]}_{\text{Lemma 5.12}} + \sum_{k=1}^{K} \underbrace{\left[ \mathbb{E}[\tilde{r}_k^\top q^*] - \mathbb{E}[\tilde{r}_k^\top q_k] \right]}_{\text{Lemma 5.11}}$$

$$\leqslant \tilde{\mathcal{O}}\big(\sqrt{\mathcal{N}SAH^3K} + S^2AH^3\big) + SAH\sqrt{\frac{(K-1)}{2}\ln(\frac{2}{\delta})} + SAH + 2(1 + \sqrt{L_\delta})(SAH + 4\sqrt{\mathcal{C}}\sqrt{6SAHK})$$

$$\leqslant \tilde{\mathcal{O}}\big(\sqrt{\mathcal{N}SAH^3K} + S^2AH^3\big) + SAH\sqrt{\frac{(K-1)}{2}\ln(\frac{2}{\delta})} + 3(1 + \sqrt{L_\delta})(SAH + 4\sqrt{\mathcal{C}}\sqrt{6SAHK})$$

### D.1.2. VIOLATION BOUND PROOF

**Stochastic setting.** Recall the definition of stochastic Violation:

$$\text{Violation}(K) = \sum_{k=1}^{K} \left[ V^{\pi_k}(\bar{d}, p) \right]^+ = \sum_{k=1}^{K} \underbrace{\left[ V^{\pi_k}(\bar{d}, p) - V^{\pi_k}(\tilde{d}_k, \tilde{p}_k) \right]^+}_{\text{Estimation Error}} + \sum_{k=1}^{K} \underbrace{\left[ V^{\pi_k}(\tilde{d}_k, \tilde{p}_k) \right]^+}_{\text{Optimization Error}}$$

Similarly, we first use Lemma 5.10 to bound "Estimation Error" term under the stochastic setting. Next, we will deal with the optimization error term by Lemma 5.13:

$$\sum_{k=1}^{K} \left[ V^{\pi_k}(\tilde{d}_k, \tilde{p}_k) \right]^+ = \sum_{k=1}^{K} \left[ \mathbb{E}[\tilde{d}_k^\top q_k] \right]^+ \leqslant 16(1 + \sqrt{L_\delta})\sqrt{\mathcal{C}}\sqrt{6SAHK}\ln\left( K + 8\sqrt{\mathcal{C}}\left( \frac{\sqrt{6SAHK}}{SAH} \right) + 2 \right)$$

Hence, the whole stochastic Violation is bounded as:

$$\text{Violation}(K) = \sum_{k=1}^{K} \left[ V^{\pi_k}(\bar{d}, p) \right]^+ = \sum_{k=1}^{K} \underbrace{\left[ V^{\pi_k}(\bar{d}, p) - V^{\pi_k}(\tilde{d}_k, \tilde{p}_k) \right]^+}_{\text{Lemma 5.10}} + \sum_{k=1}^{K} \underbrace{\left[ V^{\pi_k}(\tilde{d}_k, \tilde{p}_k) \right]^+}_{\text{Lemma 5.13}}$$

$$\leqslant \tilde{\mathcal{O}}\big(\sqrt{\mathcal{N}SAH^3K} + S^2AH^3\big) + 16(1 + \sqrt{L_\delta})\sqrt{\mathcal{C}}\sqrt{6SAHK}\ln\left( K + 8\sqrt{\mathcal{C}}\left( \frac{\sqrt{6SAHK}}{SAH} \right) + 2 \right)$$

**Adversarial setting.** When we deal with adversarial constraint, by the definition:

$$\text{Violation}(K) = \sum_{k=1}^{K} \left[ V^{\pi_k}(d_k, p) \right]^+ = \sum_{k=1}^{K} \underbrace{\left[ V^{\pi_k}(d_k, p) - V^{\pi_k}(d_k, \tilde{p}_k) \right]^+}_{\text{Estimation Error}} + \sum_{k=1}^{K} \underbrace{\left[ V^{\pi_k}(d_k, \tilde{p}_k) \right]^+}_{\text{Optimization Error}}$$

In this situation, we proved an additional estimation error bound in Lemma 5.10 with adversarial case and with Lemma 5.13, the following bound can be obtained:

$$\text{Violation}(K) = \sum_{k=1}^{K} \left[ V^{\pi_k}(d_k, p) \right]^+ = \sum_{k=1}^{K} \underbrace{\left[ V^{\pi_k}(d_k, p) - V^{\pi_k}(d_k, \tilde{p}_k) \right]^+}_{\text{Lemma 5.10}} + \sum_{k=1}^{K} \underbrace{\left[ V^{\pi_k}(d_k, \tilde{p}_k) \right]^+}_{\text{Lemma 5.13}}$$

$$\leqslant \tilde{\mathcal{O}}\big(\sqrt{\mathcal{N}SAH^3K} + S^2AH^3\big) + 16(1 + \sqrt{L_\delta})\sqrt{\mathcal{C}}\sqrt{6SAHK}\ln\left( K + 8\sqrt{\mathcal{C}}\left( \frac{\sqrt{6SAHK}}{SAH} \right) + 2 \right)$$

### D.2. Proof of Remark 5.2

If we fix the reward and constraint, where $\tilde{r}_k = \tilde{r}_{k-1}, \tilde{d}_k = \tilde{d}_{k-1}$, then we have the following relationship adapted from Lemma 5.9:

$$\sqrt{\sum_{k=1}^{K} \|\nabla_k - \nabla_{k-1}\|^2} \leqslant \alpha\sqrt{\sum_{k=1}^{K} \Phi'(\lambda_k)\tilde{d}_k - \Phi'(\lambda_{k-1})\tilde{d}_{k-1}}$$

$$\leqslant \alpha(1 + \sqrt{L_\delta})\Phi'(\lambda_K)\sqrt{2SAHK}$$

**Tighter Bound Analysis.** Similar to proof in Appendix C.4 we can obtain:

$$\Phi(\lambda_K) + \alpha \sum_{k=1}^{K} [\tilde{r}_k^\top q^* - \tilde{r}_k^\top q_k] \leqslant 3.5\sqrt{\mathcal{C}} \left( \frac{\Phi'(\lambda_K)\sqrt{2SAHK}}{2SAH} \right)$$

$$\exp(\beta\lambda_K) - 1 + \alpha \sum_{k=1}^{K} [\tilde{r}_k^\top q^* - \tilde{r}_k^\top q_k] \leqslant \beta \exp(\beta\lambda_K) \cdot 3.5\sqrt{\mathcal{C}} \left( \frac{\sqrt{2SAHK}}{2SAH} \right)$$

$$\alpha \sum_{k=1}^{K} [\tilde{r}_k^\top q^* - \tilde{r}_k^\top q_k] \leqslant \exp(\beta\lambda_K) \left( \beta \cdot 3.5\sqrt{\mathcal{C}} \left( \frac{\sqrt{2SAHK}}{2SAH} \right) - 1 \right) + 1$$

$$\sum_{k=1}^{K} [\tilde{r}_k^\top q^* - \tilde{r}_k^\top q_k] \leqslant \exp(\beta\lambda_K) \left( \frac{\beta \cdot 3.5\sqrt{\mathcal{C}} \left( \frac{\sqrt{2SAHK}}{2SAH} \right)}{\alpha} - \frac{1}{\alpha} \right) + \frac{1}{\alpha}$$

$$\leqslant 2(1 + \sqrt{L_\delta})SAH$$

where the last equality obtained by $\alpha = \frac{1}{2(1+\sqrt{L_\delta})SAH}, \beta \leqslant \frac{2SAH}{3.5\sqrt{\mathcal{C}}\sqrt{2SAHK}}$. Now, we clearly prove a $\mathcal{O}(1)$ bound for $\sum_{k=1}^{K} [\tilde{r}_k^\top q^* - \tilde{r}_k^\top q_k]$.

# E. Useful Lemmas

**Lemma E.1** (Lemma E.15 of (Dann et al., 2017)). *Consider two MDPs $\mathcal{M}_1 = (\mathcal{S}, \mathcal{A}, \{p_h^1\}_{h=1}^H, \{r_h^1\}_{h=1}^H)$ and $\mathcal{M}_2 = (\mathcal{S}, \mathcal{A}, \{p_h^2\}_{h=1}^H, \{r_h^2\}_{h=1}^H)$. For any policy $\pi$ and $s, h$, the following relation holds.*

$$V_h^\pi(s; r^1, p^1) - V_h^\pi(s; r^2, p^2)$$
$$= \mathbb{E}\left[ \sum_{h'=h}^{H} r_h^1(s_h, a_h) - r_h^2(s_h, a_h) + (p_h^1 - p_h^2)(\cdot \mid s_h, a_h)V_{h+1}^\pi(\cdot; r^1, p^1) \mid s_h = s, \pi, p^2 \right] \quad (37)$$

*where $(p_h^1 - p_h^2)(\cdot \mid s_h, a_h)V_{h+1}^\pi(\cdot; r^1, p^1) = \sum_{s \in \mathcal{S}}(p_h^1 - p_h^2)(s' \mid s_h, a_h)V_{h+1}^\pi(s'; r^1, p^1)$.*

**Lemma E.2** (Lemma D.5 of (Liu et al., 2021b)). *With probability at least $1 - \delta$,*

$$\sum_{k=1}^{K} \sum_{h=1}^{H} \sum_{(s,a)} \frac{q_h^{\pi_k}(s, a)}{\sqrt{n_h^{k-1}(s, a) \vee 1}} \leqslant 6HSA + 2H\sqrt{SAK} + 2HSA\ln K + 5\ln\frac{2HK}{\delta}$$

$$\sum_{k=1}^{K} \sum_{h=1}^{H} \sum_{(s,a)} \frac{q_h^{\pi_k}(s, a)}{n_h^{k-1}(s, a) \vee 1} \leqslant 4HSA + 2HSA\ln K + 4\ln\frac{2HK}{\delta} \quad (38)$$

*where $q_h^{\pi_k}(s, a) = \Pr(s_h = s, a_h = a \mid s_1, \pi_k, p)$.*

**Lemma E.3** (Lemma 8 of (Jin et al., 2020)). *Conditioned on the good event $G$, for all $(s, a, h, s', k) \in \mathcal{S} \times \mathcal{A} \times [H] \times \mathcal{S} \times [K]$, there exists constants $C_1, C_2 > 0$ for which we have for all $\tilde{p}^k \in B_k$ that*

$$|(p_h - \tilde{p}_h^k)(s' \mid s, a)| \leqslant C_1\sqrt{\frac{p_h(s, a)L_\delta^p}{n_h^{k-1}(s, a) \vee 1}} + \frac{C_2 L_\delta^p}{n_h^{k-1}(s, a) \vee 1}.$$

The following lemma is Lemma 10 of (Chen & Luo, 2021) with a boundedness constant $C$ for the reward function $r_k$.

**Lemma E.4** (Lemma 10 of (Chen & Luo, 2021)). *Let $r_k$ be an arbitrary function such that $r_{k,h}(s, a) \in [-C, C]$ for all $(s, a, h, k) \in \mathcal{S} \times \mathcal{A} \times [H] \times [K]$. If the true transition kernel $p$ satisfies $p \in B_k$, then for any $\tilde{p}^k \in B_k$ we have*

$$\sum_{k=1}^{K} \left| \mathbb{E}\left[ \sum_{h=1}^{H} (p_h - \tilde{p}_h^k)(\cdot \mid s_h, a_h)(V_{h+1}^{\pi_k}(\cdot; r_k, p) - V_{h+1}^{\pi_k}(\cdot; r_k, \tilde{p}^k)) \mid s_1, \pi_k, p \right] \right| = \tilde{\mathcal{O}}(CH^3S^2A).$$

**Lemma E.5** (Lemma 4 of (Chen & Luo, 2021)). *For any reward function $r$, policy $\pi$, transition kernel $p$,*

$$\text{Var}\left( \sum_{h=1}^{H} r_h(s_h, a_h) \mid s_1, \pi, p \right) \geqslant \mathbb{E}\left[ \sum_{h=1}^{H} \mathbb{V}_h(s_h, a_h; \pi, p) \mid s_1, \pi, p \right]. \quad (39)$$

**Lemma E.6** (Estimation Error for $p$)**.** *Let $\tilde{p}_k$ denote the transition kernel in the candidate set $B_k$, and let $\ell_k$ be an arbitrary function with $\ell_{k,h}(s,a) \in [-C, C]$ for all $(s, a, h, k) \in \mathcal{S} \times \mathcal{A} \times [H] \times [K]$ and some $C > 0$. Then, conditioned on the good event $G$, the estimation error on the transition kernel can be bounded as follows:*

$$\sum_{k=1}^{K} |V^{\pi_k}(\ell_k, p) - V^{\pi_k}(\ell_k, \tilde{p}_k)| \leqslant \tilde{\mathcal{O}}(C\sqrt{\mathcal{N}SAH^3K} + CS^2AH^3).$$

*Proof.* By Lemma E.1, the desired term can be rewritten as

$$\sum_{k=1}^{K} |V^{\pi_k}(\ell_k, p) - V^{\pi_k}(\ell_k, \tilde{p}^k)| = \sum_{k=1}^{K} \left| \mathbb{E}\left[ \sum_{h=1}^{H} (p_h - \tilde{p}_h^k)(\cdot \mid s_h, a_h) V_{h+1}^{\pi_k}(\cdot; \ell_k, \tilde{p}^k) \mid s_1, \pi_k, p \right] \right|$$

$$\leqslant \underbrace{\sum_{k=1}^{K} \left| \mathbb{E}\left[ \sum_{h=1}^{H} (p_h - \tilde{p}_h^k)(\cdot \mid s_h, a_h) V_{h+1}^{\pi_k}(\cdot; \ell_k, p) \mid s_1, \pi_k, p \right] \right|}_{\text{Term (I)}}$$

$$+ \underbrace{\sum_{k=1}^{K} \left| \mathbb{E}\left[ \sum_{h=1}^{H} (p_h - \tilde{p}_h^k)(\cdot \mid s_h, a_h)(V_{h+1}^{\pi_k}(\cdot; \ell_k, p) - V_{h+1}^{\pi_k}(\cdot; \ell_k, \tilde{p}^k)) \mid s_1, \pi_k, p \right] \right|}_{\text{Term (II)}}$$

where we use the short-hand notation $p_h(\cdot \mid s, a)V^{\pi}(\cdot; \ell_k, p) = \sum_{s' \in \mathcal{S}} p_h(s' \mid s, a)V^{\pi}(s'; \ell_k, p)$. Under the good event $G$, we have

$$|(p_h - \tilde{p}_h^k)(s' \mid s, a)| \leqslant C_1 \sqrt{\frac{p_h(s,a)L_\delta^p}{n_h^{k-1}(s,a) \vee 1}} + \frac{C_2 L_\delta^p}{n_h^{k-1}(s,a) \vee 1} \tag{40}$$

due to Lemma E.3. Applying this, Term (I) can be written as

$$\text{Term (I)} = \sum_{k=1}^{K} \left| \mathbb{E}\left[ \sum_{h=1}^{H} (p_h - \tilde{p}_h^k)(\cdot \mid s_h, a_h) V_{h+1}^{\pi_k}(\cdot; \ell_k, p) \mid s_1, \pi_k, p \right] \right|$$

$$= \sum_{k=1}^{K} \left| \mathbb{E}\left[ \sum_{h=1}^{H} \sum_{s'} (p_h - \tilde{p}_h^k)(s' \mid s_h, a_h) V_{h+1}^{\pi_k}(s'; \ell_k, p) \mid s_1, \pi_k, p \right] \right|$$

$$= \sum_{k=1}^{K} \left| \mathbb{E}\left[ \sum_{h=1}^{H} \sum_{s'} (p_h - \tilde{p}_h^k)(s' \mid s_h, a_h)(V_{h+1}^{\pi_k}(s'; \ell_k, p) - \mathbb{E}_{s'' \sim p_h(\cdot | s_h, a_h)} V_{h+1}^{\pi_k}(s''; \ell_k, p)) \mid s_1, \pi_k, p \right] \right|$$

$$\leqslant \sum_{k=1}^{K} \mathbb{E}\left[ \sum_{h=1}^{H} \sum_{s'} |(p_h - \tilde{p}_h^k)(s' \mid s_h, a_h)| \left| V_{h+1}^{\pi_k}(s'; \ell_k, p) - \mathbb{E}_{s'' \sim p_h(\cdot | s_h, a_h)} V_{h+1}^{\pi_k}(s''; \ell_k, p) \right| \mid s_1, \pi_k, p \right]$$

$$\leqslant \underbrace{\sum_{k=1}^{K} \mathbb{E}\left[ \sum_{h=1}^{H} \sum_{s'} C_1 \sqrt{\frac{p_h(s' \mid s_h, a_h)L_\delta^p}{n_h^{k-1}(s_h, a_h) \vee 1}} \left| V_{h+1}^{\pi_k}(s'; \ell_k, p) - \mathbb{E}_{s'' \sim p_h(\cdot | s_h, a_h)} V_{h+1}^{\pi_k}(s''; \ell_k, p) \right| \mid s_1, \pi_k, p \right]}_{\text{Term (I-a)}}$$

$$+ \underbrace{\sum_{k=1}^{K} \mathbb{E}\left[ \sum_{h=1}^{H} \sum_{s'} \frac{C_2 L_\delta^p}{n_h^{k-1}(s_h, a_h) \vee 1} \left| V_{h+1}^{\pi_k}(s'; \ell_k, p) - \mathbb{E}_{s'' \sim p_h(\cdot | s_h, a_h)} V_{h+1}^{\pi_k}(s''; \ell_k, p) \right| \mid s_1, \pi_k, p \right]}_{\text{Term (I-b)}}$$

where the third equality follows from $\sum_{s'} (p_h - \tilde{p}_h^k)(s' \mid s, a)\mathbb{E}_{s'' \sim p_h(\cdot | s_h, a_h)} V_{h+1}^{\pi_k}(s''; \ell_k, p) = 0$, and the last inequality is due to (40). Note that the expectations can be expressed by occupancy measures. Furthermore, in Term (I-a), we can replace

$\sum_{s'}$ with $\sum_{s':p_h(s'|s,a)>0}$. Then this can be rewritten as

Term (I-a)

$$= C_1\sqrt{L_\delta^p} \sum_{k=1}^K \mathbb{E}\left[ \sum_{h=1}^H \sum_{s':p_h(s'|s,a)>0} \sqrt{\frac{p_h(s'\mid s_h,a_h)}{n_h^{k-1}(s_h,a_h)\vee 1}} \left|V_{h+1}^{\pi_k}(s';\ell_k,p) - \mathbb{E}_{s''\sim p_h(\cdot|s_h,a_h)}V_{h+1}^{\pi_k}(s'';\ell_k,p)\right| \mid s_1,\pi_k,p\right]$$

$$= C_1\sqrt{L_\delta^p} \sum_{k=1}^K \mathbb{E}\left[ \sum_{h=1}^H \sum_{s':p_h(s'|s,a)>0} \sqrt{\frac{p_h(s'\mid s_h,a_h)(V_{h+1}^{\pi_k}(s;\ell_k,p) - \mathbb{E}_{s''\sim p_h(\cdot|s_h,a_h)}V_{h+1}^{\pi_k}(s'';\ell_k,p))^2}{n_h^{k-1}(s_h,a_h)\vee 1}} \mid s_1,\pi_k,p\right]$$

$$= C_1\sqrt{L_\delta^p} \sum_{k=1}^K \sum_{(s,a,h)} q_h^{\pi_k}(s,a;p) \sum_{s':p_h(s'|s,a)>0} \sqrt{\frac{p_h(s'\mid s,a)(V_{h+1}^{\pi_k}(s;\ell_k,p) - \mathbb{E}_{s''\sim p_h(\cdot|s,a)}V_{h+1}^{\pi_k}(s'';\ell_k,p))^2}{n_h^{k-1}(s,a)\vee 1}}$$

$$\leqslant C_1\sqrt{L_\delta^p}\sqrt{\sum_{k=1}^K \sum_{(s,a,h)} q_h^{\pi_k}(s,a;p) \sum_{s':p_h(s'|s,a)>0} p_h(s'\mid s,a)(V_{h+1}^{\pi_k}(s;\ell_k,p) - \mathbb{E}_{s''\sim p_h(\cdot|s,a)}V_{h+1}^{\pi_k}(s'';\ell_k,p))^2}$$

$$\times \sqrt{\sum_{k=1}^K \sum_{(s,a,h)} \sum_{s':p_h(s'|s,a)>0} \frac{q_h^{\pi_k}(s,a;p)}{n_h^{k-1}(s,a)\vee 1}}$$

where the inequality follows from the Cauchuy-Schwarz inequality. Here,

$$\sum_{s':p_h(s'|s,a)>0} p_h(s'\mid s,a)(V_{h+1}^{\pi_k}(s;\ell_k,p) - \mathbb{E}_{s''\sim p_h(\cdot|s,a)}V_{h+1}^{\pi_k}(s'';\ell_k,p))^2 = \mathbb{V}_h(s,a;\pi_k,p).$$

Then, Term (I-a) is upper bounded as

$$\text{Term (I-a)} \leqslant C_1\sqrt{L_\delta^p}\sqrt{\sum_{k=1}^K \sum_{(s,a,h)} q_h^{\pi_k}(s,a;p)\mathbb{V}_h(s,a;\pi_k,p)} \times \sqrt{\sum_{k=1}^K \sum_{(s,a,h)} \sum_{s':p_h(s'|s,a)>0} \frac{q_h^{\pi_k}(s,a;p)}{n_h^{k-1}(s,a)\vee 1}}$$

$$\leqslant C_1\sqrt{L_\delta^p}\sqrt{\sum_{k=1}^K \mathbb{E}\left[\sum_{h=1}^H \mathbb{V}_h(s_h,a_h;\pi_k,p) \mid s_1,\pi_k,p\right]} \times \sqrt{\mathcal{N}\sum_{k=1}^K \sum_{(s,a,h)} \frac{q_h^{\pi_k}(s,a;p)}{n_h^{k-1}(s,a)\vee 1}}.$$

By Lemma E.5, we have $\mathbb{E}\left[\sum_{h=1}^H \mathbb{V}_h(s_h,a_h;\pi_k,p) \mid s_1,\pi_k,p\right] \leqslant \text{Var}\left(\sum_{h=1}^H \ell_{k,h}(s_h,a_h) \mid s_1,\pi_k,p\right)$. Since $\sum_{h=1}^H \ell_{k,h}(s_h,a_h) \in [-CH,CH]$ almost surely, we have $\text{Var}\left(\sum_{h=1}^H \ell_{k,h}(s_h,a_h) \mid s_1,\pi_k,p\right) \leqslant C^2H^2$. Furthermore, we can bound the later term with Lemma E.2. Then it follows that

$$\text{Term (I-a)} = \tilde{\mathcal{O}}(C\sqrt{KH^2} \times \sqrt{\mathcal{N}SAH}) = \tilde{\mathcal{O}}(C\sqrt{\mathcal{N}SAH^3K}).$$

To bound Term (I-b), under the good event $G$, we have $\left|V_{h+1}^{\pi_k}(s';\ell_k,p) - \mathbb{E}_{s''\sim p_h(\cdot|s_h,a_h)}V_{h+1}^{\pi_k}(s'';\ell_k,p)\right| \leqslant 2CH$. By Lemma E.2,

$$\text{Term (I-b)} \leqslant 2CHS \sum_{k=1}^K \sum_{h=1}^H \mathbb{E}\left[\frac{C_2L_\delta^p}{n_h^{k-1}(s_h,a_h)\vee 1} \mid s_1,\pi_k,p\right] = \tilde{\mathcal{O}}\left(CH^2S^2A\right).$$

Then we have

$$\text{Term (I)} = \text{Term (I-a)} + \text{Term (I-b)} = \tilde{\mathcal{O}}\left(C\sqrt{\mathcal{N}SAH^3K} + CH^2S^2A\right).$$

By Lemma E.4, we can bound Term (II) as

$$\text{Term (II)} = \tilde{\mathcal{O}}(CH^3S^2A).$$

Finally, we have

$$\sum_{k=1}^K |V^{\pi_k}(\ell_k,p) - V^{\pi_k}(\ell_k,\tilde{p}^k)| \leqslant \text{Term (I)} + \text{Term (II)} = \tilde{\mathcal{O}}(C\sqrt{\mathcal{N}SAH^3K} + CH^3S^2A).$$

$\square$

