# OpenReview forum: "An Optimistic Algorithm for online CMDPS with Anytime Adversarial Constraints"
_ICML.cc/2025/Conference — ICML 2025 poster_

### Official Review · Reviewer_7ukS · 2025-03-11

**Overall Recommendation:** 4

**Summary:**

The paper considers the online learning problems in the constrained MDP problems. The objective is to maximize the reward while satisfying the constraints, where the system state transits according to an underlying MDP. The problem is formulated as a finite horizon episodic setting with $H$ periods in each episode. The main goal of the paper is to develop an on-policy learning algorithm to achieve a $O(\sqrt{K})$ regret upper bound on the objective and the constraint violation, where $K$ denotes the number of episodes.

The main algorithm of the paper is based on a primal-dual framework. The optimal policy for the CMDP problems can be formulated as an LP. However, the LP cannot be directly solved since the problem parameters are unknown. Therefore, the paper considers the Lagrangian dual formulation of the LP and updates the primal variable, which is the occupancy measure, and the dual variable in an online manner. To be specific, the paper employs an optimistic mirror descent algorithm to update the primal variable occupancy measure, while using the online gradient descent algorithm to update the dual variable. At each step, the newly observed data is again collected to refine the estimation of the problem parameters, which will be used in the update of the next step.

The paper shows that by properly picking the parameters in the mirror descent algorithms that update the primal and dual, the main algorithm of the paper is able to achieve a regret bound of $O(\sqrt{K})$ over the total reward and the constraint violations. The paper further shows that when the reward and cost functions are deterministic, then the algorithm would achieve an $O(1)$ bound on the total reward, since the primal variable is updated using optimistic mirror descent, however, the constraint violation bound turns out to still be at the order of $O(\sqrt{K})$.

One theoretical contribution claimed by the paper is that their result does not require the slater's condition to be satisfied, which is usually adopted to bound the range of the dual variable by the existing algorithms. Therefore, even if there is no guarantee of the existence of a strictly feasible solution, the algorithm in the paper still works.

**Claims And Evidence:**

The claim looks convincing to me.

**Essential References Not Discussed:**

The algorithm developed in the paper is based on the LP formulation, where the primal-dual algorithmic framework is widely developed to solve it. However, primal-based algorithms have also been developed to solve the LP formulation or the CMDP problems in general. For example, the primal-based algorithm has been developed in [1], [2], [3], [4], and [5]. In particular, the paper [6] develops resolving primal LP methods to solve the CMDP and achieves the first instance-dependent $O(1/\epsilon)$ sample complexity. I think these papers are worth mentioning in the literature.

References:

[1]. Yongshuai Liu, Jiaxin Ding, and Xin Liu. Ipo: Interior-point policy optimization under constraints. In Proceedings of the AAAI conference on artificial intelligence, volume 34, pages 4940–4947, 2020.

[2]. Yinlam Chow, Mohammad Ghavamzadeh, Lucas Janson, and Marco Pavone. Risk-constrained reinforcement learning with percentile risk criteria. Journal of Machine Learning Research, 18(167):1–51, 2018.

[3]. Yinlam Chow, Ofir Nachum, Aleksandra Faust, Edgar Duenez-Guzman, and Mohammad Ghavamzadeh. Lyapunov-based safe policy optimization for continuous control. arXiv preprint arXiv:1901.10031, 2019.

[4]. Gal Dalal, Krishnamurthy Dvijotham, Matej Vecerik, Todd Hester, Cosmin Paduraru, and Yuval Tassa. Safe exploration in continuous action spaces. arXiv preprint arXiv:1801.08757, 2018.

[5]. Tengyu Xu, Yingbin Liang, and Guanghui Lan. Crpo: A new approach for safe reinforcement learning with convergence guarantee. In International Conference on Machine Learning, pages 11480–11491. PMLR, 2021.

[6]. Jiang, Jiashuo, and Yinyu Ye. "Achieving $\tilde {O}(1/\epsilon) $ Sample Complexity for Constrained Markov Decision Process." NeurIPS, 2024.

**Experimental Designs Or Analyses:**

There is no numerical experiment.

**Methods And Evaluation Criteria:**

The proposed algorithm comes with theoretical guarantees. However, there is no numerical validation of the algorithm.

**Other Comments Or Suggestions:**

Please refer to my questions below.

**Other Strengths And Weaknesses:**

Strength:

The paper develops a primal-dual algorithm to solve the CMDP problems. Using the optimistic mirror descent to update the primal variable is an interesting idea and the benefits is that when the reward and cost functions are deterministic, the bound over the gap of objective values can be improved to $O(1)$ (though the constraint violation gap is still $O(\sqrt{T})$). In the general case, the algorithm enjoys an $O(\sqrt{K})$ bound over the gap of objective values and constraint violations.

Weakness:

1. The $O(\sqrt{K})$ bound (or the equivalent $O(1/\epsilon^2)$ sample complexity) is now quite standard in the literature and has been developed in many previous works. Though the result can be improved to $O(1)$ when the reward and cost functions are deterministic, the bound over the constraint violation is still $O(\sqrt{K})$.

2. It is claimed that the paper considers a stronger formulation of the constraint, as discussed in line 60. However, it turns out that the problem can still be formulated as LP, which is the same as the formulation induced by the weaker formulation of the constraint. The algorithm is also developed to solve the LP in an online primal-dual manner. Therefore, I am not sure whether the stronger anytime constraint formulation would introduce any difference here. I have asked a question for further clarification from the authors.

3. The primal-dual framework has been widely developed, see for example Efroni et al., 2020. The main difference of the algorithm in the paper seems to be that optimistic mirror descent has been adopted to update the primal variables. However, the only benefits seems to be that when the reward and cost functions are deterministic, the bound over the gap of objective values can be improved to $O(1)$. But the bound on the constraint violation is still $O(\sqrt{K})$. So the benefits of using optimistic mirror descent is not immediately clear to me.

**Questions For Authors:**

1. It is claimed that the paper considers a stronger formulation of the constraint, as discussed in line 60. However, it turns out that the problem can still be formulated as LP with decision variables being the occupancy measure, which is the same as the formulation induced by the weaker formulation of the constraint. The algorithm is also developed to solve the LP in an online primal-dual manner. Therefore, I am not sure whether the stronger anytime constraint formulation would introduce any difference here. Could you please provide further clarification on this?

2. Could you provide more explanation on the benefits of using an optimistic mirror descent to update the primal variable? I understand that one potential benefit is that when the reward and cost functions are deterministic, the bound over the gap of objective values can be improved to $O(1)$. But the bound on the constraint violation is still $O(\sqrt{K})$.

3. One of the theoretical contribution is that the algorithm does not require the slater's condition to be satisfied. However, previous work, for example, Efroni et al. (2020), requires this condition to develop an upper bound on the dual variable, which is useful for scaling the range of the dual variable to a unit box. However, I think even if the slater's condition does not hold, there must be some other way to upper bound the dual variable, or we just assume a given upper bound over the dual variable. So it seems to me that removing the Slater's condition is not a big part. Please correct me if I am wrong about this.

4. Could you please comment on the practical performance of your algorithm, since there is no numerical experiment in the paper?

**Relation To Broader Scientific Literature:**

The paper develops primal-dual algorithms to study the constrained MDP problems. The algorithm is based on a LP formulation of the CMDP problems. The new part of the algorithm is to utilize the optimistic mirror descent algorithm to update the primal decision variable.  The paper shows that such a design could achieve a better bound over the objective value when the reward and cost functions are deterministic.

**Theoretical Claims:**

The proof looks correct to me.

---

> ### Author Rebuttal · Authors · 2025-04-01
>
> We appreciate the valuable comments provided by the reviewer. We address the reviewer's questions as follows.
> >**Response to Essential References Not Discussed**
>
> We will explicitly cite and discuss these references in our final revised manuscript, clearly positioning our primal-dual contributions against this closely related primal-only approach.
>
> >**Response to Weakness and Question**
>
> >**W1: Regarding Standard Result** We would like to mention that although the $\tilde{O}(\sqrt{K})$ regret for reward is standard, it is the optimal order under unknown reward, other approaches can not achieve this result even under a deterministic reward. For the constraint violation, achieving a $\tilde{O}(\sqrt{K})$ bound in the adversarial constraint setting is not standard and is extremely difficult and nontrivial. We are the first to establish such a result in the anytime constraints setting.
>
> >**W2: Difference of Stronger and Weaker constraint** LP only serves as a general framework for solving optimization problems, it is well-known that the CMDP problem is equivalent to an LP problem. So LP is used as a standard tool for optimizing the inner problem at each step, however the primal-dual formulation and the Lyapunov function may be different. In other words, the theoretical results established in our paper are due to the carefully constructed Lyapunov function and LP is formulation is due to the natural of the CMDP problem. Please let us know if further explanation is needed. We will also revise the manuscript to make these points clearer.
>
> >**W3: Benefits of using OMD** It is indeed true that the primal-dual method is a central tool for solving CMDP problems, primarily because CMDPs can be formulated as linear programs and are essentially constrained optimization problems. However, the primal-dual framework is just one part of the overall solution strategy. There remain many open challenges in CMDPs due to their numerous variants—such as soft vs. hard constraints, adversarial constraints, and both model-based and model-free formulations. In our work, the achieved regret bound of $\tilde{\mathcal{O}}(\sqrt{K})$ matches the known lower bound and is optimal in the case of unknown rewards. In contrast, other approaches cannot achieve the $\mathcal{O}(1)$ reward bound even under deterministic rewards. As for constraint violation, obtaining a $\tilde{\mathcal{O}}(\sqrt{K})$ bound under adversarial constraints is highly non-trivial and, to our knowledge, has not been shown before in the {anytime constraint} setting. Our work is the first to establish such a result. The use of optimistic mirror descent (OMD) in our algorithm enables us to bound the regret through a sequence of one-step gradients of the surrogate objective function $f$ across $K$ episodes, as shown in Lemma 5.7. This approach, combined with a carefully designed Lyapunov function, ensures both low regret and strong constraint satisfaction. In summary, OMD is critical not only for improving convergence in the deterministic setting, but also for controlling the gap between the observed and optimal performance in each episode. Together with our Lyapunov-based dual control mechanism, it enables a unified and near-optimal treatment of both stochastic and adversarial CMDPs. We will revise the manuscript to better emphasize this connection and clarify the contributions beyond the use of OMD.
>
> >**Q1: Regarding Standard Result** Besides the response in Weakness-(2), algorithms developed under weaker constraint formulations are not directly applicable to stronger formulations. This is because weaker constraints allow for cancellation of constraint violations during the learning process (see lines 56–65 in the main paper), a property that does not hold in the stronger constraint setting.
>
> >**Q2: Benefits of using OMD** Please see the response in Weakness-(3). Please let us know if further explanation is needed
>
> >**Q3: Removing Slater’s condition** Due to response length limitations, we have addressed this question under Reviewer 7EBa. Kindly refer to the response to Weakness-(1) in that section.
>
> >**Q4: Experiment Result** Due to response length limitations, we have addressed this question under Reviewer 7EBa. Kindly refer to the response to Weakness-(3) in that section.
>
> We sincerely thank the reviewer for the helpful feedback. If our response satisfactorily addresses your concerns, we would greatly appreciate your consideration in raising the evaluation score. We're happy to clarify any further questions during the discussion.

---

### Official Review · Reviewer_7EBa · 2025-03-12

**Overall Recommendation:** 4

**Summary:**

This paper introduces the Optimistic Mirror Descent Primal-Dual (OMDPD) algorithm, a novel approach for online constrained Markov decision processes (CMDPs) with anytime adversarial constraints. Unlike prior methods that assume known safe policies or rely on Slater’s condition, OMDPD achieves optimal regret $\tilde{O}(\sqrt{K})$ and strong constraint violation bounds $\tilde{O}(\sqrt{K})$ even in dynamic and adversarial settings. The algorithm leverages optimistic estimates, online mirror descent, and adaptive dual updates to balance exploration and constraint satisfaction. If accurate reward and transition estimates are available (via a generative model), the regret can further improve to O(1). OMDPD surpasses existing CMDP approaches by handling both stochastic and adversarial constraints, making it highly relevant for applications in autonomous systems, robotics, and cybersecurity. ## update after rebuttal

**Claims And Evidence:**

The paper provides a thorough theoretical analysis and compares its results against various existing CMDP algorithms. The findings demonstrate that the proposed method achieves sublinear regret and sublinear constraint violation under adversarial settings, effectively validating its theoretical claims.

**Essential References Not Discussed:**

None.

**Experimental Designs Or Analyses:**

No experiment results. While the paper provides strong theoretical guarantees, it lacks empirical experiments to validate the practical performance of OMDPD in real-world CMDP scenarios. Benchmarking against existing algorithms on simulated or real datasets would strengthen its applicability.

**Methods And Evaluation Criteria:**

The paper introduces the Optimistic Mirror Descent Primal-Dual (OMDPD) algorithm, which combines optimistic online mirror descent (OMD) with a primal-dual framework to handle adversarial constraints in CMDPs. The method leverages optimistic estimates for transition kernels, rewards, and costs, using confidence sets and Lyapunov-based updates to regulate constraint violations dynamically. The evaluation is based on two key theoretical criteria: regret minimization and strong constraint violation bounds.

**Other Comments Or Suggestions:**

None.

**Other Strengths And Weaknesses:**

Strengths:
+ The paper establishes that OMDPD achieves optimal regret $\tilde{O}(\sqrt{K})$  and strong constraint violation bounds $\tilde{O}(\sqrt{K})$, ensuring both efficiency and safety in stochastic and adversarial settings. These bounds improve upon prior approaches that only handle stochastic constraints or allow weaker forms of violation.
+ OMDPD provides a generalized approach that works under both stochastic and adversarial rewards/costs, bridging the gap between these two settings.
+ The theoretical results show that if accurate estimates of rewards and transitions are available ( through a generative model), the regret bound can be further improved to O(1). This highlights the algorithm’s adaptability and potential for achieving near-optimal learning in CMDPs with additional information.
+ This paper uses refined error bounds(Lemma 5.9) for estimating transition kernels, rewards, and costs in CMDPs. By leveraging a Bellman-type law of total variance, it improves upon Lemma 29 of Efroni et al. (2020)[1] by a factor of $\tilde{O}(\sqrt{H})$, enabling more precise value estimation and better learning efficiency.

Weaknesses:
+ The paper claims that removing Slater’s condition is an advancement, as it eliminates the need for a strictly feasible policy. However, in many practical scenarios, a near-feasible solution is often available, making this contribution less impactful than suggested.
+ The proposed algorithm is developed under the assumption that reward and cost functions are linear, which may limit its applicability. It remains unclear whether this approach can generalize to a broader class of convex CMDP settings where rewards and costs exhibit nonlinear dependencies on state-action pairs. Many real-world problems, such as risk-sensitive decision-making or energy management, involve complex, non-convex constraints that may not align with the assumptions made in the theoretical analysis. A discussion on potential extensions to general convex or non-convex CMDPs would strengthen the paper’s contributions and applicability.
+ While the paper provides strong theoretical guarantees, it lacks empirical experiments to validate the practical performance of OMDPD in real-world CMDP scenarios. Benchmarking against existing algorithms on simulated or real datasets would strengthen its applicability.

[1] Efroni, Yonathan, Shie Mannor, and Matteo Pirotta. "Exploration-exploitation in constrained mdps." arXiv preprint arXiv:2003.02189 (2020)

**Questions For Authors:**

Please refer to the Other Strengths And Weaknesses part.

**Relation To Broader Scientific Literature:**

This paper contributes to the broader literature on constrained reinforcement learning (RL), online convex optimization, and adversarial learning by introducing a unified framework for CMDPs with both stochastic and adversarial constraints. By leveraging optimistic mirror descent and primal-dual methods, the proposed OMDPD algorithm achieves optimal regret and strong constraint violation bounds, advancing research on safe decision-making in dynamic and adversarial environments with applications in autonomous systems, robotics, and cybersecurity.

**Theoretical Claims:**

The proof of Theorem 5.1 is well-structured, and Figure 1 effectively illustrates the relationships between Lemmas 5.7–5.11. The detailed proof steps in the appendix are logically sound, clearly presented, and follow a rigorous progression.

---

> ### Author Rebuttal · Authors · 2025-04-01
>
> We appreciate the valuable comments provided by the reviewer. We address the reviewer's questions as follows.
> >**Response to Weakness**
>
> >**W1: Removing Slater’s condition** While it is true that in some practical scenarios, a near-feasible solution may exist, it doesn't make any changes to our results. A relaxed assumption is always desired for theoretical analysis. We would like to emphasize that, from a theoretical perspective, we achieve the best result without the need for the Slater's condition—an important and nontrivial contribution. The reason is that some existing methods, [1, 2, 3, 4, 5] rely on Slater’s condition with a slackness parameter $\rho$ , which quantifies how well the condition is satisfied. Their final regret and constraint violation bounds depend directly on this parameter, and when $\rho$ is small, these bounds can degrade significantly. Moreover, some algorithms([1, 3]) need to know the slackness $\rho$ which is not practical.
>
> [1] "Optimal Strong Regret and Violation in Constrained MDPs via Policy Optimization."
>
> [2] "Online learning in CMDPs: Handling stochastic and adversarial constraints."
>
> [3] "Cancellation-free regret bounds for lagrangian approaches in constrained markov decision processes."
>
> [4] "Learning adversarial mdps with stochastic hard constraints."
>
> [5] "Best-of-Both-Worlds Policy Optimization for CMDPs with Bandit Feedback."
>
> >**W2: Convex CMDP applicability** We would like to clarify that our algorithm is not inherently restricted to linear reward and cost functions. In fact, it can be naturally extended to a broader class of convex CMDP settings where the reward and cost functions are general convex functions of the occupancy measure $ q $.  Specifically, if we define the reward and cost functions as $ h_k(q) $ and $ g_k(q) $, respectively, the general optimization problem becomes:
> $$
> \max_{q \in \mathcal{Q}} \ \frac{1}{K} \sum_{k=1}^K h_k(q) \quad \text{s.t.} \quad g_k(q) \leq 0.
> $$
> To ensure theoretical guarantees, we require the following basic conditions:
> (1) The reward is generated stochastically; The cost can be generated either stochastically or adversarially;
> (2) The potential function $f$ used in the updates is 1-strongly convex;
> (3) The occupancy measure lies in a compact convex set(Fact 5.3);
> (4) The reward $h_k$ and cost $g_k$ functions are convex and Lipschitz continuous with respect to the occupancy measure $ q $:$ |h_k(q_1) - h_k(q_2)| \leq C_1 \|q_1 - q_2\|_2, \quad |g_k(q_1) - g_k(q_2)| \leq C_2 \|q_1 - q_2\|_2.$
> Under these conditions, the core structure of our algorithm remains applicable. We will provide the proof in our final version.
>
> >**W3: Experiment Results** We implemented our algorithm on a synthetic CMDP with the following settings: $\mathcal{S} = 5$, $\mathcal{A} = 3$, and $H = 5$. For the adversarial constraint, the cost function was randomly selected from a fixed cost set with values in the range $[-1,1]$. After running the algorithm for $K = 3{,}000$ episodes, we report the cumulative constraint violation in the table below:
>
> | Episode               | K=1   | K=500   | K=1000  | K=1500  | K=2000  | K=2500  | K=3000  |
> |-----------------------|-------|---------|---------|---------|---------|---------|---------|
> | Constraint Violation  | 12.75 | 757.07  | 882.14  | 957.41  | 1012.13 | 1055.41 | 1091.21 |
>
> It is clear that a sublinear regret for the constraint violation is achieved.
>
> We sincerely thank the reviewer for the helpful feedback. If our response satisfactorily addresses your concerns, we would greatly appreciate your consideration in raising the evaluation score. We're happy to clarify any further questions during the discussion.

---

> > ### Comment · Reviewer_7EBa · 2025-04-02
> >
> > Thanks for the response. The authors' explanations have thoroughly addressed my previous questions, particularly regarding convex CMDP applicability, and the experiment results show strong performance under adversarial constraints. I am willing to raise my score.

---

> > > ### Author Response · Authors · 2025-04-02
> > >
> > > Thank you for your acknowledgment and for increasing the score of our paper,  we truly appreciate it!

---

### Official Review · Reviewer_zWwP · 2025-03-24

**Overall Recommendation:** 4

**Summary:**

The paper studies the online learning problem for episodic constrained Markov decision processes, where the constraint functions are either stochastic or adversarial, and the transition function is unknown. A key technique the authors introduce is a surrogate objective function for the policy optimization step. The authors propose an optimistic mirror descent primal-dual algorithm: (1) the primal step updates the occupancy measure via the standard optimistic gradient step for the surrogate objective function; (2) the dual step performs the standard dual ascent step. In both stochastic and adversarial settings, the authors prove that the standard regret and the strong constraint violation (only counts non-negative violations) are sublinear in the number of episodes, without assuming Slater's condition and knowing a strictly feasible policy.

**Claims And Evidence:**

(1) A key quantity of the main result is $\mathcal{C}$ which is used throughout the proof. Since it is not well-defined for the KL divergence, the main result can be vacuous.

(2) For the adversarial cost case, the authors assume the feasibility of all constraints, which is not the standard adversarial loss setting.

(3) In the regret analysis, the comparison policy is a nearly feasible policy instead of an optimal policy. In this sense, the regret bounds can be suboptimal.

**Essential References Not Discussed:**

To the best of my knowledge, essential references have been discussed.

**Experimental Designs Or Analyses:**

The authors didn't provide experimental results.

**Methods And Evaluation Criteria:**

The proposed primal-dual method seems appropriate for the problem, but its conditions for correct performance characterization remain unclear.

**Other Comments Or Suggestions:**

It would be helpful if the authors could compare different techniques in the literature that lead to varying regret guarantees and highlighted the technical challenges in achieving strong regret or constraint violation bounds in the adversarial/stochastic settings.

**Other Strengths And Weaknesses:**

Other weaknesses on clarity:

(1) The violation analysis of adversarial constraints in Theorem 5.12 follows the same analysis. This suggests that the adversarial setting is not inherently more challenging than the stochastic setting. It would be helpful if the authors could clarify the technical challenges for analyzing adversarial constraints.

(2) The authors apply the optimistic online mirror descent to a surrogate objective function for the primal update, achieving strong constraint violation bounds. It would be helpful if the author could clarify why this is the case for the strong constraint violation bounds, not strong regret bounds. The optimistic gradient method was used in the following papers to improve convergence. It would be helpful if the authors could clarify the challenges of obtaining strong regret and constraint violation bounds.

- ReLOAD: Reinforcement Learning with Optimistic Ascent-Descent for Last-Iterate Convergence in Constrained MDPs

- Last-Iterate Convergent Policy Gradient Primal-Dual Methods for Constrained MDPs

(3) A key step to bound the violation of adversarial constraints is to use the drift-plus-penalty framework for (23). It would be helpful if the authors could provide a formal proof of it for the CMDP setting?

**Questions For Authors:**

(1) What is the definition of anytime adversarial constraints?

(2) What is the definition of $\mathcal{N}$ in Theorem 5.1?

(3) Is the optimal policy in Theorem 5.6 episode-dependent?

(4) How do accurate estimates of rewards and transitions improve regret bounds?

**Relation To Broader Scientific Literature:**

This work is of interest to the broader safe reinforcement learning community.

**Theoretical Claims:**

I went through the proof sketch in Section 5, but I didn't check the proofs. The authors provide a proof sketch for the adversarial and stochastic settings. Their key difference is the violation analysis. It is not clear the reason why both cases can use the same comparison policy in Lemma 5.6.

---

> ### Author Rebuttal · Authors · 2025-04-01
>
> We appreciate the valuable comments provided by the reviewer. We address the reviewer's questions as follows.
>
> >**C1: Bounded $\mathcal{C}$** In our paper, we assume that $U$ is a 1-strongly convex function. When $U(q)$ is the entropy function, the resulting Bregman divergence is the KL divergence, which lacks a guaranteed upper bound. A common remedy is assuming a bound on the KL divergence [1] or introduce a probability mixing step to avoid boundary issues [1][2]. In both cases, the constant $\mathcal{C}$ is measure-dependent but independent of the time horizon $K$, so our algorithm's performance bound remains valid.
>
> [1] arxiv.org/abs/1908.00305
>
> [2] arxiv.org/abs/1311.1869
>
> >**C2: Adversarial setting**  We clarify that the adversarial loss setting in our paper follows the standard formulation widely studied in constrained online convex optimization ([1,2]). The reviewer may be referring to a different setting, where constraints are satisfied only in expectation over time ([3]). In contrast, our work focuses on the stricter anytime constraint setting, requiring constraint satisfaction in every episode. To our knowledge, ours is the first to achieve a sublinear $\tilde{O}(\sqrt{K})$ constraint violation under this setting, which we hope will inspire further research.
>
> [1] arxiv.org/abs/2310.18955
>
> [2] "Online convex optimization with hard constraints: Towards the best of two worlds and beyond."
>
> [3] "Online learning in CMDPs: Handling stochastic and adversarial constraints."
>
> >**C3: Regrading near optimal policy** The comparison policy is the optimal policy for the CMDP problem (3),  we decomposite the regret analysis into two error terms in Eq.(24). This is the standard definition of regret.
>
> >**Methods Criteria**  We clarify that our theoretical guarantees rely on the following key assumptions: (1) without Slater condition (a main contribution); (2) stochastic rewards, stochastic/adversarial costs; (3) the potential function is 1-strongly convex (due to our Lyapunov choice); (4) the occupancy measure is a compact convex set; and (5) reward/cost functions are convex and Lipschitz in the occupancy measure.
>
> >**Theoretical Claims and Experiment** We would like to clarify that we use a single notation $\pi^*$ to denote the optimal policy under both stochastic and adversarial constraint settings for simplicity, although the optimal policies in these two cases are generally different, which theoretical analysis applies to both scenarios. Kindly refer to the response to Weakness-(3) in Reviewer 7EBa to check the experiment results.
>
>
> >**W1: Challenging of adversarial setting** While our algorithm is designed for the adversarial constraint, it also applies to the stochastic constraint setting. This is one of the main contributions of our work—the proposed algorithm provides a unified framework that achieves near-optimal performance in both settings, without requiring prior knowledge of the environment type. Compared with existing approaches, reaching this level of generality and optimality requires careful design of the surrogate objective function, which ensures that unsafe actions are avoided while maintaining regret bounds through the use of the OMD algorithm. Our carefully constructed Lyapunov function plays a key role in guaranteeing these theoretical properties.
>
> >**W2:** We emphasize that minimizing regret and enforcing strong constraint satisfaction are different objectives. Our paper primarily addresses adversarial constraint violations. While both [1][2] adopt OMD, the key novelty of our work lies in Eq.(23), which provides an upper bound on the sum of exponential dual variable and cumulative estimated regret. This result is enabled by Lyapunov function (Eq.(20)), which incorporates both the reward and cost via rectified transformation. The theoretical contribution doesn't arise solely from using the OMD update. Its advantage is that it allows us to bound the episode-wise difference $\sum_{k=1}^K \left(f_k(q_k) - f_k(q^*)\right)$ in a clean way. Since [1, 2] do not adopt a similar Lyapunov-based design, they are unable to provide strong bounds on constraint violations under adversarial costs. Moreover, from the drift analysis, we show that the dual variable $\lambda_K$ can be controlled by bounding its exponential transformation.
>
> >**W3:** The proof of Eq.(23) is shown in the proof of Lemma 5.8(Appendix C.3, line 816-851).
>
> >**Q1:** The anytime adversarial constraint assumes that the cost functions are chosen adversarially at each episode, and the agent is required to satisfy the constraints in anytime.
>
> >**Q2:** $\mathcal{N}$ is maximum number of non-zero transition probabilities
>
> >**Q3:**  The optimal policy is not episode-dependent, defined in Eq.(3).
>
> >**Q4: $O(1)$ bound** The improvement can be achieved due to our design of the Lyapunov function and the OMD component. Following Lemma 5.8, the term(I),(III) will disapper when reward is deterministic, which will benefit the bound.

---

### Decision · Program_Chairs · 2025-05-01

**Decision:**

Accept (poster)

**Comment:**

This paper tackles constraint MDP with adversarial loss and adversarial constraints (the constraint is represented in the form of another cost function).  It improves previous work by removing the regret dependency on the constant involved in the Slater's condition, and also allows for adversarial constraints.  The former improvement comes from the use of a surrogate objective function that has appeared in previous work for full-information settings.  Overall, the technique used in this work is a natural combination of existing techniques, and turns out to give a new bound in the considered setting. I recommend for acceptance.

Reviewer zWwP pointed out that the constant $\mathcal{C}$ in Theorem 5.1 is undetermined.  Although the proposed solution is reasonable and standard, please make it clear in the revised version.